 **eLIFE**

# BRAF inhibitors suppress apoptosis through off-target inhibition of JNK signaling

Harina Vin[1†], Sandra S Ojeda[1†], Grace Ching[1†], Marco L Leung[1,2], Vida Chitsazzadeh[1,2], David W Dwyer[1], Charles H Adelmann[1], Monica Restrepo[1], Kristen N Richards[1,3], Larissa R Stewart[1,3], Lili Du[1], Scarlett B Ferguson[4], Deepavali Chakravarti[2,5], Karin Ehrenreiter[6], Manuela Baccarini[6], Rosamaria Ruggieri[7], Jonathan L Curry[8], Kevin B Kim[9], Ana M Ciurea[3], Madeleine Duvic[2,3], Victor G Prieto[8], Stephen E Ullrich[1,2], Kevin N Dalby[4], Elsa R Flores[2,5], Kenneth Y Tsai[1,2,3]*

[1]Department of Immunology, University of Texas MD Anderson Cancer Center, Houston, United States; [2]Graduate School of Biomedical Sciences at Houston, University of Texas, Houston, United States; [3]Department of Dermatology, University of Texas MD Anderson Cancer Center, Houston, United States; [4]Division of Medicinal Chemistry, College of Pharmacy, University of Texas, Austin, United States; [5]Department of Biochemistry and Molecular Biology, University of Texas MD Anderson Cancer Center, Houston, United States; [6]Max F Perutz Laboratories, Vienna, Austria; [7]Center for Oncology and Cell Biology, Feinstein Institute for Medical Research, Manhasset, United States; [8]Department of Pathology, University of Texas MD Anderson Cancer Center, Houston, United States; [9]Department of Melanoma Medical Oncology, University of Texas MD Anderson Cancer Center, Houston, United States

*For correspondence: kytsai@mdanderson.org

†These authors contributed equally to this work

Competing interests: The authors declare that no competing interests exist.

**Abstract** Vemurafenib and dabrafenib selectively inhibit the v-Raf murine sarcoma viral oncogene homolog B1 (BRAF) kinase, resulting in high response rates and increased survival in melanoma. Approximately 22% of individuals treated with vemurafenib develop cutaneous squamous cell carcinoma (cSCC) during therapy. The prevailing explanation for this is drug-induced paradoxical ERK activation, resulting in hyperproliferation. Here we show an unexpected and novel effect of vemurafenib/PLX4720 in suppressing apoptosis through the inhibition of multiple off-target kinases upstream of c-Jun N-terminal kinase (JNK), principally ZAK. JNK signaling is suppressed in multiple contexts, including in cSCC of vemurafenib-treated patients, as well as in mice. Expression of a mutant ZAK that cannot be inhibited reverses the suppression of JNK activation and apoptosis. Our results implicate suppression of JNK-dependent apoptosis as a significant, independent mechanism that cooperates with paradoxical ERK activation to induce cSCC, suggesting broad implications for understanding toxicities associated with BRAF inhibitors and for their use in combination therapies.

## Introduction

BRAF inhibitors (BRAFi) have revolutionized the treatment of melanoma (*Flaherty et al., 2010*; *Chapman et al., 2011*; *Sosman et al., 2012*; *Falchook et al., 2012*; *Hauschild et al., 2012*; *Long et al., 2012*). Their clinical use is associated with the development of keratinocytic tumors including cSCC (*Flaherty et al., 2010*; *Chapman et al., 2011*; *Sosman et al., 2012*; *Hauschild et al., 2012*; *Falchook et al., 2012*;

**eLife digest** Over 50% of melanomas, a highly lethal form of skin cancer, carry mutations in a gene called BRAF. The BRAF gene encodes an enzyme that helps to regulate the proliferation of cells, but mutations in this gene lead to the excessive proliferation that is seen in cancer. Clinical trials have shown that a drug called vemurafenib can be used to treat patients who carry the mutated BRAF genes and go on to develop melanoma, but around one fifth of these patients developed another type of skin cancer called cSCC (cutaneous squamous cell carcinoma).

The cSCC tumors often develop in areas where the sun has damaged the patient's skin, and it is thought that their growth is then accelerated by vemurafenib activating another enzyme, ERK, which causes the excessive proliferation of skin cells. Vin et al. have now found that vemurafenib might also cause cSCC tumors by blocking another signaling pathway. The experiments were performed in human cells and also in mice, and the results were then verified in human cSCC samples.

Cells that are exposed to UV radiation usually die, but when treated with vemurafenib, some 70% of the cells that would have died instead survived. The stress from the UV radiation activates the JNK signaling pathway, which causes the irradiated cells to die. However, Vin et al. found that cSCC cells had very low levels of JNK signaling because treatment with vemurafenib had the unintended effect of inhibiting three enzymes that are needed to fully activate the JNK signaling pathway.

Vin et al. estimate that suppression of JNK signaling and cell death is responsible for about 17.6 to 40% of the effect on cSCC growth seen in melanoma patients, with activation of the ERK pathway accounting for the rest. These unexpected findings suggest that combining vemurafenib treatment with radiation or chemotherapy should be done with caution as these effects could affect their efficacy. It also suggests that future drugs should be designed in a way that avoids these types of effects by making sure they do not inhibit important 'off-target' enzymes.

*Long et al., 2012*). Mechanistic studies of this have centered on paradoxical ERK activation, which is most evident in *BRAF*-wild-type, *RAS*-mutant cells, as the primary mechanism (*Karreth et al., 2009*; *Halaban et al., 2010*; *Hatzivassiliou et al., 2010*; *Heidorn et al., 2010*; *Poulikakos et al., 2010*). This is supported by the findings that *RAS* mutations are significantly enriched in cSCC arising in patients treated with vemurafenib relative to sporadic cSCC (*Oberholzer et al., 2011*; *Su et al., 2012*), and by the low rate of cSCC in patients treated with combined BRAFi and MEK inhibitor (MEKi) (*Flaherty et al., 2012*). In one model, drug binding relieves the autoinhibition of BRAF whereupon it is recruited to the membrane by activated RAS and dimerizes with CRAF, driving MEK-dependent ERK activation (*Heidorn et al., 2010*). Other studies show ERK hyperactivation resulting from drug-induced CRAF transactivation (*Hatzivassiliou et al., 2010*; *Poulikakos et al., 2010*) and modulation of RAS spatiotemporal dynamics (*Cho et al., 2012*). Inhibitor-induced KSR1-BRAF dimers modulate the activity of ERK (*McKay et al., 2011*) and also affect MEK signaling by activating KSR1 kinase activity (*Brennan et al., 2011*; *Hu et al., 2011*). These models all highlight the importance of CRAF in driving MEK-dependent hyperactivation of ERK.

Because of the rapid development of these cSCC on BRAFi therapy and the enrichment for *RAS* mutations, pre-existing genetic lesions are likely present prior to therapy, which are then 'unmasked' following initiation of BRAFi therapy. The fact that many arise in sun-damaged skin suggests that prior chronic UV exposure is an important predisposing event (*Su et al., 2012*).

We instead hypothesized that vemurafenib and PLX4720 could also affect the susceptibility of cells to apoptosis and in so doing, contribute to the acceleration of tumor development. We studied the acute ultraviolet radiation (UVR) response because this is the most important environmental risk factor in the development of skin cancer and because many BRAFi-induced cSCC arise in sun-damaged areas (*Su et al., 2012*). PLX4720 and vemurafenib share structural features (*Tsai et al., 2008*; *Bollag et al., 2010*) and have similar activities, as is the case in our studies.

## Results

### BRAFi suppress stress-induced, JNK-dependent apoptosis
We performed our initial studies using cSCC (SRB1, SRB12, COLO16) and keratinocyte (HaCaT) cell lines. Cells treated with 1 kJ/m$^2$ of UVB (FS40 lamp) undergo apoptosis within 24 hr (*Figure 1A–D*).

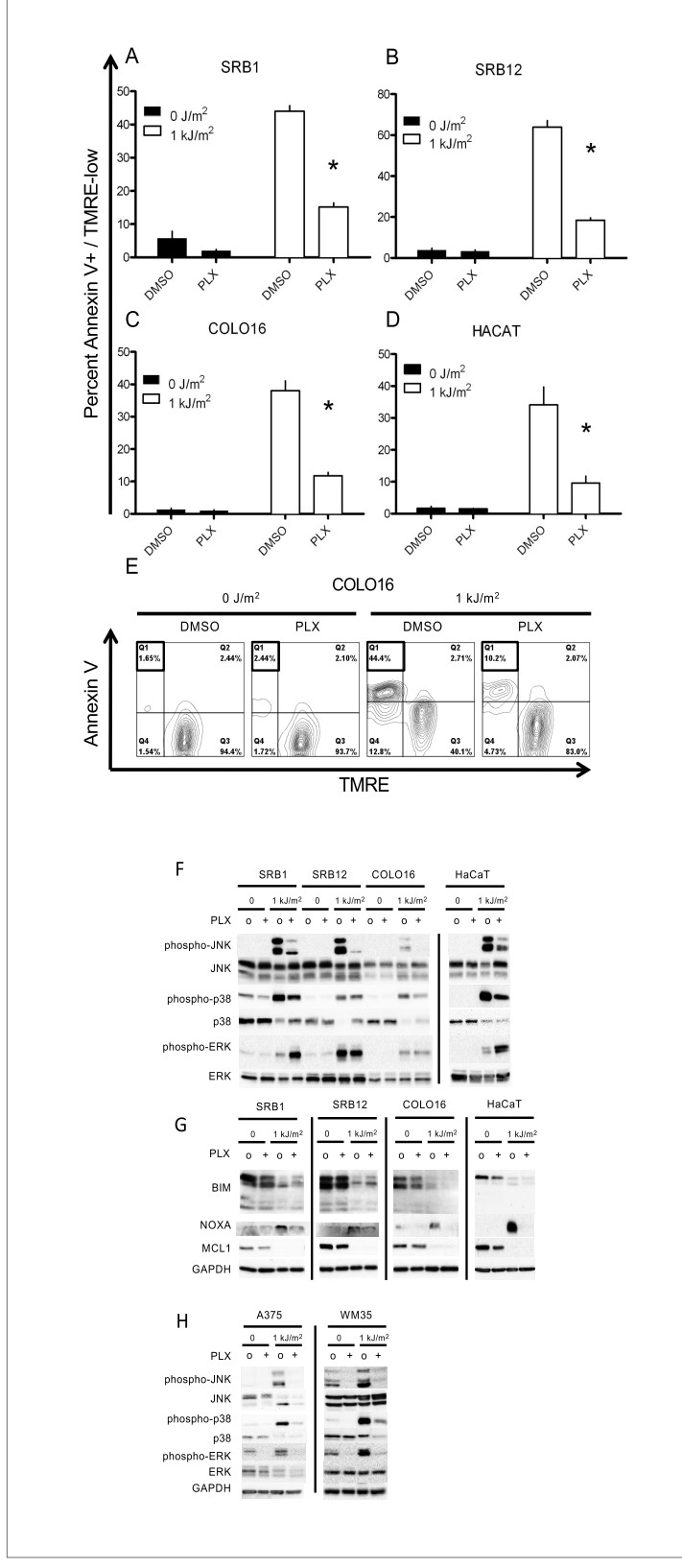

**Figure 1**. PLX4720 suppresses UV-induced apoptosis. The cSCC and HaCaT cell lines were either unirradiated or irradiated with 1 kJ/m² of UVB in the absence ('o', 1:2000 DMSO) or presence ('+') of 1 µM PLX4720 and isolated for FACS analysis and protein extracts 24 hr later. (**A**) SRB1, (**B**) SRB12, (**C**) COLO16, and (**D**) HaCaT cells show
*Figure 1. Continued on next page*

*Figure 1. Continued*

at least 70% suppression of apoptosis in the presence of PLX4720 as measured by FACS for Annexin V+, TMRE-low cells (n = 6 for each cell line, '\*' denotes statistical significance at p<0.05). (**E**) A representative FACS plot for COLO16 is shown. Annexin V+, TMRE-low cells are contained in the upper left quadrant (boxed), which was significantly populated in UV-irradiated cells, but not in the absence of UV, or in the presence of PLX4720. (**F**) Western blots probed for the MAP kinases demonstrated strong phospho-JNK and phospho-p38 induction following irradiation and significant suppression by PLX4720. Phospho-ERK was induced following irradiation, and at 24 hr, paradoxical hyperactivation in the presence of PLX4720 was observed in SRB1 and HaCaT cells. (**G**) Western blots showed that BIM was not upregulated in these *BRAF*-wild-type cells, consistent with intact ERK signaling. MCL1 was downregulated by irradiation and not modulated by PLX4720, whereas NOXA expression was strongly induced in irradiated cells and suppressed by PLX4720. (**H**) Western blots of *BRAF*$^{V600E}$ melanoma cell lines, A375 and WM35, demonstrated suppression of UV-mediated induction of phospho-JNK and phospho-p38 by PLX4720 at 24 hr. As expected, phospho-ERK is shut down in PLX4720-treated cells.

The following figure supplements are available for figure 1:

**Figure supplement 1**. PLX4720 potently suppresses apoptosis in cSCC, HaCaT cell lines, and NHEK cells.

**Figure supplement 2**. PLX4720 suppresses doxorubicin-induced JNK activation and apoptosis in cSCC and HaCaT cell lines.

**Figure supplement 3**. PLX4720 does not confer a proliferative advantage to cSCC and HaCaT cell lines.

Surprisingly, this apoptosis was suppressed by at least 70% in cells concomitantly treated with 1 µM PLX4720 (*Figure 1A–D*) compared to control DMSO-treated cells as measured by FACS for Annexin V+; TMRE (tetramethylrhodamine)-low cells (*Figure 1E*, *Figure 1—figure supplement 1A–C*). Similar results were obtained using doxorubicin as the inducer of apoptosis, and similar suppression of apoptosis was obtained using 1 µM PLX4720 in all cells (*Figure 1—figure supplement 2A,B*). Importantly, these cells have no oncogenic *RAS* or *BRAF* mutations (*Table 1*), and PLX4720 conferred no significant proliferative advantage to the tested cells (*Figure 1—figure supplement 3*) even when used at concentrations that inhibit the proliferation of *BRAF*$^{V600E}$ melanoma cell lines (*Tsai et al., 2008*).

Because the p38 and JNK stress-activated MAP kinases are well-established critical mediators of UV-induced apoptosis (*Derijard et al., 1994*; *Chen et al., 1996*; *Tournier et al., 2000*; *Hildesheim et al., 2004*), we explored the status of JNK and p38 activation by assessing phospho-JNK and phospho-p38 levels by Western blot (*Figure 1F*). Phospho-JNK levels in particular were highly upregulated upon UV irradiation and were significantly suppressed by treatment post-radiation with 1 µM PLX4720 in cSCC and HaCaT cell lines (*Figure 1F*). Similar effects were seen with 1 µM vemurafenib (data not shown) and in cells stressed with doxorubicin (*Figure 1—figure supplement 2C*). Importantly, ERK signaling remained intact, as evidenced both by the paradoxical activation of ERK (upregulation of phospho-ERK) and by the failure to upregulate BIM levels (*Figure 1F,G*). This pro-apoptotic BCL2 family member is upregulated by inhibition of ERK signaling (*Collins et al., 2005*) and in *BRAF*$^{V600E}$ melanoma cells treated with vemurafenib (*Paraiso et al., 2011*). Since NOXA is a downstream effector of UV-induced apoptosis (*Naik et al., 2007*), we examined its expression and found that NOXA expression is induced by UV irradiation and suppressed by PLX4720 in all cell lines (*Figure 1G*), suggesting that inhibition of NOXA expression may be a mechanism of PLX4720-induced suppression of apoptosis. Finally, we examined the expression of the antiapoptotic BCL2 family member MCL1 because it is downregulated by UV exposure (*Figure 1G*), but as previously reported (*Paraiso et al., 2011*), unaffected by PLX4720 (*Figure 1G*).

To test the generality of these effects in cells in which ERK activity is suppressed by BRAFi, we extended our analysis to the BRAF$^{V600E}$ melanoma cells A375 and WM35. As expected, phospho-ERK expression was strongly suppressed by PLX4720 (*Figure 1H*). Phospho-JNK and phospho-p38 were significantly upregulated following UV-irradiation (*Figure 1H*), showing that signaling to JNK and p38 is intact in *BRAF*$^{V600E}$ melanoma cells. Here again, there was significant suppression of both phospho-p38 and phospho-JNK induction by PLX4720 (*Figure 1H*), and similar effects were seen with vemurafenib (data not shown).

We next examined the responses of primary normal human epidermal keratinocytes (NHEKs) to vemurafenib. UV-induced apoptosis was significantly suppressed (approximately 70%) by vemurafenib

**Table 1.** Lack of *BRAF* and *RAS* mutations in cSCC and HaCaT cell lines

| | |
|---|---|
| ALK_F1174LIV_T3520CAG | ALK_F1245C_T3734G |
| ALK_F1245VI_T3733GA | BRAF_G464EVA_G1391ATC |
| BRAF_G466R_G1396CA | BRAF_K601E_A1801G |
| BRAF_V600EAG_T1799ACG_F | CTNNB1_S45APT_T133GCA |
| CTNNB1_T41APS_A121GCS | EGFR_Y813C_A2438G |
| GNAS_R201SC_C601AT | KRAS_G12SRC_G34ACT |
| KRAS_Q61EKX_C181GAT | MET_H1112_A3335GT |
| MET_Y1248HD_T3742CG | PIK3CA_A1046V_C3137T |
| PIK3CA_C420R_T1258C | PIK3CA_E110K_G328A |
| PIK3CA_E418K_G1252A | PIK3CA_F909L_C2727G |
| PIK3CA_H1047RL_A3140 GT | PIK3CA_H701P_A2102C |
| PIK3CA_N345K_T1035A | PIK3CA_Q060K_C178A |
| PIK3CA_R088Q_G263A | PIK3CA_S405F_C1214T |
| TNK2_R99Q_G296A | BRAF_G466EVA_G1397ATC |
| BRAF_V600LM_G1798 TA | CTNNB1_S37APT_T109GCA |
| CTNNB1_S45CFY_C134GTA | EGFR_G719_G2155TA |
| EGFR_L858R_T2573G | EGFR_T790M_C2369T |
| EPHA3_K761N_G2283 | FGFR2_S252W_C755G |
| FOXL2_C134W_C402G | KIT_K642E_A1924G |
| KIT_R634W_C1900T | KIT_V560D_T1679A |
| KIT_V825A_T2474C | KIT_Y553N_T1657A |
| KRAS_G12DAV_G35ACT | MET_N375S_A1124G |
| NRAS_G12SRC_G34ACT | PIK3CA_E453K_G1357A |
| PIK3CA_E545AGV_A1634CGT | PIK3CA_H1047RL_A3140 GT..1. |
| PIK3CA_K111N_G333C | PIK3CA_M1043V_A3127G |
| PIK3CA_P539R_C1616G | BRAF_E586K_G1756A |
| BRAF_G469EVA_G1406ATC | CTNNB1_S33APT_T97GCA |
| CTNNB1_S37CFY_C110GTA | EGFR_L861_T2582AG |
| EGFR_T854I_C2561T | FGFR2_N549KK_T1647GA |
| FRAP_R2505P_G7514C | FRAP_S2215Y_C6644T |
| IDH2_R172MK_G515 TA | JAK2_V617F_G1849T |
| KIT_L576P_T1727C | KIT_N566D_A1696 G |
| KRAS_A146PT_G436CA | NRAS_G12DAV_G35ACT |
| KRAS_Q61HHE_A183CTG | KRAS_G13SRC_G37ACT |
| NRAS_G13DAV_G38ACT | NRAS_Q61HHQ_A183TCG |
| PDGFRA_N659Y_A1975T | PDGFRA_V561D_T1682A |
| PIK3CA_E545KQ_G1633AC | PIK3CA_H1047Y_C3139T |
| PIK3CA_Q546EK_C1636 GA | PIK3CA_Y1021HN_T3061CA |
| RET_M918T_T2753C | AKT1_G173R_G517C |
| AKT2_E17K_G49A | BRAF_G469R_G1405CA |
| BRAF_L597R_T1790G | BRAF_V600_G1800 |
| CTNNB1_G34EVA_G101ATC | EGFR_S720P_T2158C |
| GNA11_Q209LP_A626 TC | IDH1_R132CGS_C394TGA |
| IDH2_R140LQ_G419 TA | IDH2_R140W_C418T |
| IDH2_R172S_G516T | KIT_D816HNY_G2446CAT |

*Table 1. Continued on next page*

*Table 1. Continued*

| | |
|---|---|
| KIT_V559ADG_T1676CAG | KRAS_G10R_G28A |
| KRAS_Q61LPR_A182TCG | MET_H1112Y_C3334T |
| MET_M1268T_T3803C | MET_T1010I_C3029T |
| NRAS_A146T_G436A | NRAS_Q61EKX_C181GAT |
| PDGFRA_D842V_A2525T | PDGFRA_D842_G2524TA |
| PDGFRA_N659K_C1977A | PIK3CA_E542KQ_G1624AC |
| PIK3CA_G1049R_G3145C | PIK3CA_M1043I_G3129ATC |
| PIK3R1_D560Y_G1678T | PRKAG2_N488I_A1463T |
| AKT2_G175R_G523C | AKT3_G171R_G511A |
| ALK_F1174L_C3522AG | ALK_I1171N_T3512A |
| ALK_R1275QL_G3824AT | BRAF_D594GV_A1781 GT |
| CTNNB1_D32HNY_G94CAT | FBWX7_R465C_C1393T |
| FBWX7_R479QL_G1436AT | FBWX7_R505HLP_G1514ATC |
| FGFR3_G370C_G1108T | GNAQ_Q209H_A627T |
| IDH2_R140W_C419T | IDH2_R172GW_A514 GT |
| KIT_N822KNK_T2466GCA | KRAS_G13DAV_G38ACT |
| PDPK1_D527E_C1581G | PIK3CA_E542VG_A1625TG |
| PIK3CA_E545D_G1635CT | PIK3CA_T1025SA_A3073TG |
| PIK3CA_Y1021C_A3062G | PIK3R1_N564K_C1693AG |
| PRKAG1_R70Q_G209A | AKT1_E17K_G49A |
| AKT1_K179M_A536T | BRAF_V600EAG_T1799ACG_R |
| CDK4_R24C_C70T | CDK4_R24H_G71A |
| CTNNB1_D32AGT_A95CGV | FBWX7_R465HL_G1394AT |
| FGFR3_G697C_G2089T | FGFR3_K650MT_A1949 TC |
| FGFR3_R248C_C742T | FGFR3_S371C_A1111T |
| FGFR3_Y373C_A1118G | GNAS_R201H_G602A |
| IDH1_R132HL_G395AT | KIT_N822YHD_A2464TCG |
| MET_R988C_C2962T | MET_Y1253D_T3757 G |
| NRAS_G13SRC_G37ACT | NRAS_Q61RPL_A182GCT |
| PIK3CA_Q546LPR_A1637TCG | TNK2_E346K_G1036A |
| PIK3CA_H1047RL_A3140 GT | ALK_F1245C_T3734G |

The listed gene mutations were screened by Sequenom INT16/20 panel (Characterized Cell Line Core, MD Anderson Cancer Center) and *HRAS* was sequenced by Sanger sequencing. All examined loci were wild-type in the cSCC cell lines SRB1, SRB12, COLO16, and keratinocyte cell line HaCaT. The PIK3R1_M326I_G978 polymorphism was found in the SRB12 cell line.

in these cells (*Figure 2A*, *Figure 1—figure supplement 1D*), and the UV-induced upregulation of phospho-JNK and phospho-p38 was likewise suppressed most significantly at 6 and 24 hr (*Figure 2B*). As in the cSCC and HaCaT cell lines, activation of ERK was observed following exposure to vemurafenib (*Figure 2B*). The presence of cleaved caspase-3 correlated with high levels of apoptosis in the UV-treated cells and its absence with rescue by vemurafenib at 24 hr post-irradiation (*Figure 2C*). In probing members of the BCL2 family, we found similar results to those in the cSCC and HaCaT cell lines. BIM and MCL1 were unaffected by vemurafenib but NOXA induction at 24 hr post-UV irradiation was diminished by vemurafenib (*Figure 2C*). The advantage of using primary cells is that *p53* is intact. In NHEKs, p53 is stabilized by 24 hr post-UV irradiation and this is unaffected by vemurafenib (*Figure 2C*). However, since BCL2 family members can be modulated by JNK (*Tournier et al., 2000*; *Haeusgen et al., 2011*) and p53 (*Oda et al., 2000*) in apoptosis, the inhibition of NOXA expression by PLX4720 and vemurafenib (*Figures 1G and 2C*) likely reflects *p53*-independent regulation of NOXA given that

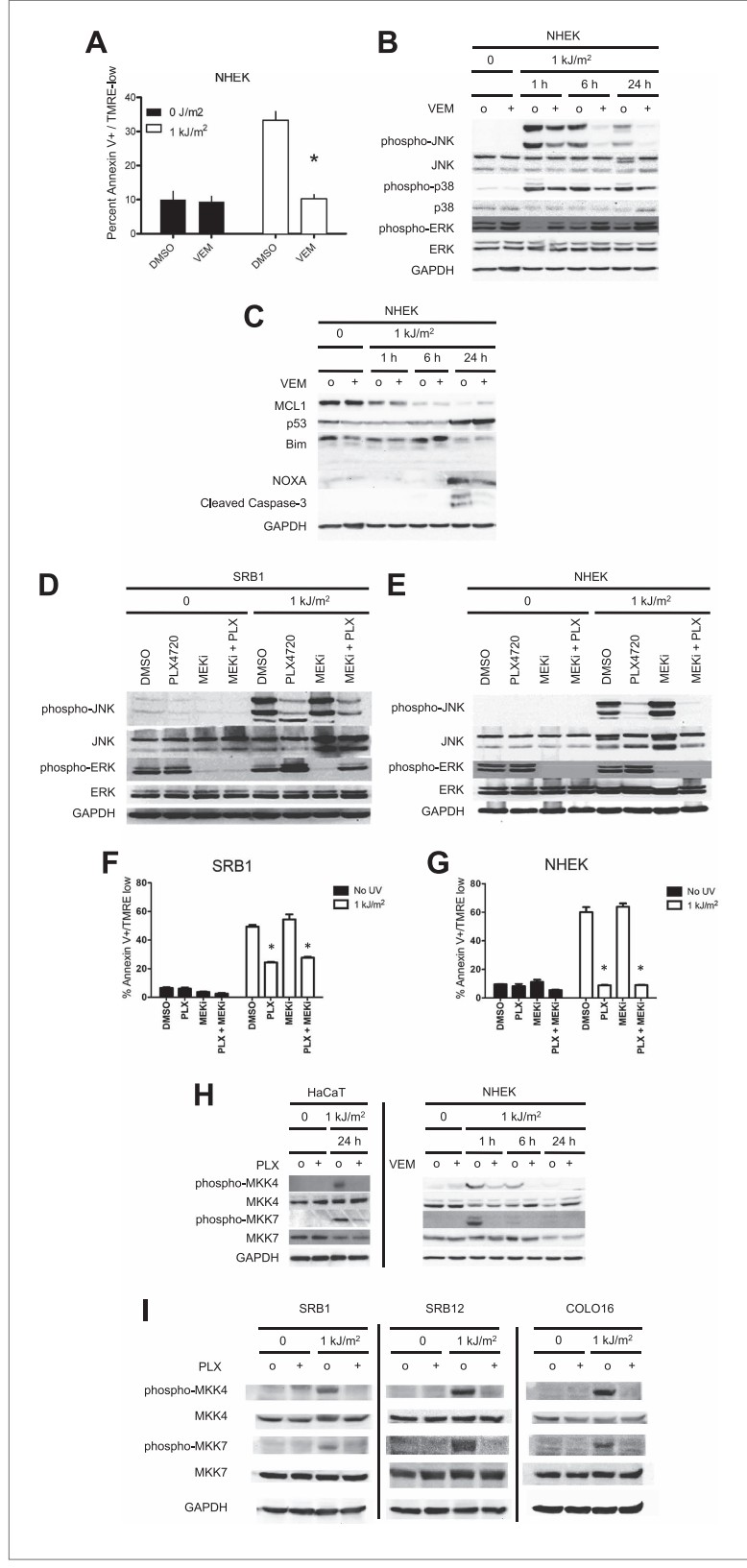

**Figure 2**. Vemurafenib and PLX4720 suppress apoptosis and JNK signaling in primary human keratinocytes and cSCC cells independently of MEK/ERK signaling. Normal human epidermal keratinocytes (NHEKs) were irradiated with 1 kJ/m² of UVB in the absence ('o', 1:2000 DMSO) or presence ('+') of 1 µM vemurafenib and isolated for FACS

*Figure 2. Continued on next page*

*Figure 2. Continued*

analysis and protein extracts 24 hr later. (**A**) Apoptosis was significantly suppressed (70%) in the presence of vemurafenib as measured by FACS for Annexin V+, TMRE-low cells (n = 6, '*' denotes statistical significance at p<0.05). (**B**) Western blot analysis showed induction of phospho-JNK and phospho-p38 within 1 hr following irradiation, which persisted for at least 24 hr and which was suppressed by vemurafenib at all time points. (**C**) MCL1 and BIM expression was not significantly modulated by vemurafenib; however, NOXA induction, which occurred at 24 hr, was reduced by vemurafenib. In these primary cells, p53 protein was stabilized by 24 hr and vemurafenib did not affect overall levels. Suppression of apoptosis, as measured by cleaved caspase-3 levels, was observed in the presence of vemurafenib-treated irradiated cells, consistent with the FACS results. To test the relevance of MEK signaling, cSCC (SRB1) and NHEK cells were irradiated with 1 kJ/m² of UVB in the absence ('o', 1:2000 DMSO) or presence of 1 μM PLX4720 singly or in combination with 0.6 μM (NHEK) or 1.2 μM (SRB1) AZD6244 (MEKi) and isolated for FACS analysis and protein extracts 24 hr later. (**D**) SRB1 and (**E**) NHEK cells showed induction of phospho-JNK at 24 hr following irradiation, by Western in the presence (lane 7) and absence (lane 5) of MEKi. The addition of MEKi to PLX4720 did not affect the suppression of JNK activation (compare lanes 6, 8) despite potent suppression of phospho-ERK. (**F**) SRB1 and (**G**) NHEK cells exhibited a strong suppression of UV-induced apoptosis by PLX4720 (Annexin V+, TMRE-low cells; n = 6, '*' denotes statistical significance at p<0.05) that was likewise unaffected by the addition of MEKi. To test whether upstream kinases in the JNK pathway were inhibited, MKK4 and MKK7 activation was probed in cells. (**H**) Both phospho-MKK4 and phospho-MKK7 were induced in HaCaT and NHEK cells following irradiation, and this was suppressed in the presence of 1 μM PLX4720 and vemurafenib, respectively. (**I**) In all cSCC cell lines, SRB1, SRB12, COLO16, phospho-MKK4 and phospho-MKK7 are strongly induced following irradiation, and this is suppressed in all lines by 1 μM PLX4720.

The following figure supplements are available for figure 2:

**Figure supplement 1**. p53 does not respond to stress in cSCC and HaCaT cell lines.

**Figure supplement 2**. BCL2 family members BCL2, BCL-XL, and BCL2A1 are not modulated by acute UV exposure or PLX4720.

---

*p53* is mutant in HaCaT (*Lehman et al., 1993*) cells, p53 is undetectable in SRB12 cells, and p53 levels do not change with radiation in SRB1, COLO16, or HaCaT cells, (*Figure 2—figure supplement 1*). PUMA, BAX, BCL2, BCL-XL, and BCL2A1 expression were unchanged following irradiation and were unchanged by PLX4720 or vemurafenib exposure (data not shown, *Figure 2—figure supplement 2*). We conclude from our results that vemurafenib and PLX4720 suppress UV-induced apoptosis by inhibiting JNK signaling and NOXA induction in *BRAF* and *RAS* WT cells.

## BRAFi suppress JNK activity through off-target inhibition of ZAK, MKK4, MAP4K5

Although BRAFi-induced JNK inhibition is observed in BRAF-WT as well as BRAF$^{V600E}$ cells (*Figure 1F,H, 2B*), with opposite effects on ERK signaling, we sought to further demonstrate that JNK inhibition and paradoxical ERK activation are independent and separable. We treated SRB1 and NHEK cells with the MEK inhibitor (MEKi) AZD6244 and PLX4720 singly and in combination with and without UV irradiation. While MEKi effectively abrogated ERK phosphorylation and activation (*Figure 2D,E*), this left PLX4720-mediated suppression of UV-induced JNK activation (*Figure 2D,E*) and apoptosis (*Figure 2F,G*) unaffected in both SRB1 and NHEK cells.

Because JNK and p38 isoforms are not significantly inhibited by PLX4720 or vemurafenib directly, (*Tsai et al., 2008*; *Bollag et al., 2010*) we probed the phosphorylation status of MKK4 and MKK7 (MAP2K7), the two proximal kinases that synergistically phosphorylate JNK and that are required for JNK activation (*Tournier et al., 2001*; *Haeusgen et al., 2011*). The phosphorylation of both MKK4 and MKK7, corresponding to their activation, was significantly upregulated in control UV-irradiated cells and inhibited by PLX4720 in all cSCC cell lines, HaCaT, and NHEK cells (*Figure 2H,I*).

We then performed a kinome screen of PLX4720 and vemurafenib against a panel of 38 kinases reported to be upstream of JNK (*Keshet and Seger, 2010*; *Haeusgen et al., 2011*) and other kinases previously tested against PLX4720 and vemurafenib (*Tsai et al., 2008*; *Bollag et al., 2010*) using a quantitative competitive binding assay (*Davis et al., 2011*) at four concentrations (50 nM, 200 nM, 1 μM, 10 μM). We extended previously reported results obtained on this platform (*Davis et al., 2011*) by testing a wider concentration range and by additionally testing vemurafenib. Reported biochemical

IC50s for vemurafenib (*Bollag et al., 2010*) and PLX4720 (*Tsai et al., 2008*) against multiple kinases including BRAF$^{V600E}$, MAP4K5, SRMS, and BRK were quantitatively similar to the estimated $K_d$, confirming the validity of this assay (*Tables 2 and 3*). We confirmed that ZAK and MKK4 (MAP2K4) have high binding affinities comparable to that of the intended target, BRAF (estimated $K_d$ below 50 nM) for both PLX4720 (*Davis et al., 2011*) and vemurafenib, and confirmed activity against MAP4K5 (*Bollag et al., 2010*) (*Tables 2 and 3*). To demonstrate an effect on activity, in vitro kinase assays were performed (*Figure 3A–C*) and revealed biochemical IC50s of 187 ± 5 nM, 460 ± 41 nM, and 354 ± 26 nM for ZAK, MKK4, and MAP4K5, respectively. All of these values are within the range of reported correspondences between binding assays and activity-based assays and with reported data (*Anastassiadis et al., 2011*; *Davis et al., 2011*). Importantly, at 1 μM vemurafenib used in our experiments, the residual activity of ZAK, MKK4, and MAP4K5 kinases, was 18.9 ± 0.5%, 29.6 ± 1.1%, and 25.7 ± 0.6%, respectively.

To examine the requirements for ZAK, MAP4K5, and MKK4 in activating JNK activation and apoptosis more directly, we performed lentiviral shRNA knockdown experiments in HaCaT cells. HaCaT cells with knockdown of ZAK ('shZAK2') showed a strong suppression of ZAK protein expression (*Figure 3—figure supplement 1A*) and of UV-induced apoptosis, showing 70% suppression of apoptosis relative to that achieved by PLX4720 in control ('SCR') cells (*Figure 3D*, *Figure 3—figure supplement 1A*). An additional clone of shRNA against ZAK ('shZAK1') showed similar results (*Figure 3—figure supplement 1B,C*), demonstrating that even greater knockdown of ZAK can account for nearly the entire effect of PLX4720 on JNK activation and apoptosis. Western blots show significant suppression of phospho-MKK4/MKK7 in shZAK2 knockdown cells (*Figure 3E*). Triple knockdown cells ('TKD') with combined shRNA knockdown of ZAK, MKK4, and MAP4K5 kinases, as confirmed by Western (*Figure 3E*, *Figure 3—figure supplement 1A*), showed comparable suppression of apoptosis to that of drug-treated control cells (*Figure 3D*, *Figure 3—figure supplement 1B*) and substantial suppression of phospho-MKK4/MKK7 induction (*Figure 3E*). Furthermore, single knockdown of MKK4 and MAP4K5 (*Figure 3—figure supplement 2A*), only partially suppresses UV-induced apoptosis or phospho-JNK induction in HaCaT cells (*Figure 3—figure supplement 2B,C*). Knockdown of ZAK alone was able to account for 91.3% of the suppression of UV-induced apoptosis in a distinct cell line, SRB1 (*Figure 3—figure supplement 3A,B*), with corresponding suppression of phospho-JNK induction (*Figure 3—figure supplement 3C*). As knockdown of ZAK alone can account for up to 93.7% of the effect of PLX4720 treatment, we conclude that the potent inhibition of JNK activation and resultant apoptosis by PLX4720 and vemurafenib is due to the off-target inhibition of ZAK principally, with smaller additional contributions from inhibition of MKK4 and MAP4K5, which abrogates the activation of the two kinases essential for JNK phosphorylation and activation: MKK4 and MKK7 (*Figures 2H–I,3D–E*, *Figure 3—figure supplement 4*). Consistent with our findings, ZAK has been shown to be critically important for JNK activation upstream of MKK4 and MKK7 (*Wang et al., 2005*) and doxorubicin-induced apoptosis (*Sauter et al., 2010*; *Wong et al., 2013*).

## Expression of gatekeeper mutant ZAK reverses BRAFi-mediated apoptosis suppression

Our biochemical and shRNA data showed that knockdown of ZAK suppressed phospho-JNK activation and apoptosis (*Figure 3D–E*, *Figure 3—figure supplements 1,3*) and that the degree of knockdown correlated with the degree of JNK and apoptosis suppression. To show that PLX4720 suppresses apoptosis primarily through direct action on ZAK in cells, we employed a chemical-genetic approach by engineering a gatekeeper mutant ZAK (T82Q). Gatekeeper mutant kinases, in which the threonine (T) is replaced by a larger amino acid, in our case glutamine (Q), are often rendered insensitive to small molecule inhibitors and are an important mechanism of drug resistance (*Daub et al., 2004*; *Whittaker et al., 2010*). We overexpressed equivalent amounts of ZAK (T82Q) wild-type ZAK (WT) in HaCaT cells (*Figure 3F*), and compared their UV responses.

Whereas ZAK (WT) cells were sensitive to PLX4720-mediated suppression of apoptosis (*Figures 1D and 3G*), drug-treated ZAK (T82Q)-expressing cells underwent 2.13-fold more apoptosis than drug-treated ZAK (WT) cells (bar 4 vs 8; p=0.005), corresponding to 76.9% of the levels of apoptosis in untreated cells (bars 3, 7 vs 8; p=0.08) (*Figure 3G*). The effects on apoptosis corresponded to higher levels of phospho-JNK, even in drug-treated cells expressing the ZAK (T82Q) mutant as compared to drug-treated ZAK (WT)-expressing cells at both 1 hr and 6 hr post-irradiation (lane 4 vs 8; *Figure 3H*). Sustained activation of JNK is necessary for apoptosis (*Tobiume et al., 2001*; *Kamata et al., 2005*; *Ventura et al., 2006*), and our results show that PLX4720-treated ZAK (T82Q)-expressing cells retain higher activation across 1–6 hr as compared to PLX4720-treated ZAK (WT) cells.

**Table 2.** Quantitative competitive binding assays reveal additional kinase targets of PLX4720

| Gene Name | Entrez gene Symbol | Percent control (50 nM) | Percent control (200 nM) | Percent control (1000 nM) | Percent control (10 μM) | Calculated estimate of IC50 (nM) | Published biochemical IC50 (nM) |
|---|---|---|---|---|---|---|---|
| ASK1 | MAP3K5 | 89 | 98 | 97 | 100 | 14,179.29 | |
| ASK2 | MAP3K6 | 94 | 100 | 100 | 100 | | |
| BLK | BLK | 91 | 78 | 32 | 1 | 446.56 | |
| **BRAF(V600E)** | **BRAF** | **38** | **19** | **3.9** | **0.1** | **32.04** | **13** |
| BRK | PTK6 | 47 | 14 | 2.4 | 0.2 | 30.38 | 130 |
| DLK | MAP3K12 | 95 | 98 | 100 | 92 | | |
| FGR | FGR | 69 | 38 | 11 | 2.5 | 153.47 | |
| HPK1 | MAP4K1 | 100 | 100 | 100 | 47 | | |
| LZK | MAP3K13 | 94 | 100 | 96 | 75 | | |
| MAP3K1 | MAP3K1 | 96 | 100 | 92 | 84 | | |
| MAP3K15 | MAP3K15 | 94 | 97 | 91 | 59 | | |
| MAP3K2 | MAP3K2 | 100 | 93 | 87 | 41 | | |
| MAP3K3 | MAP3K3 | 94 | 97 | 98 | 75 | | |
| MAP3K4 | MAP3K4 | 100 | 100 | 100 | 65 | | |
| MAP4K2 | MAP4K2 | 98 | 100 | 99 | 67 | | |
| MAP4K3 | MAP4K3 | 100 | 95 | 90 | 56 | | |
| MAP4K4 | MAP4K4 | 92 | 99 | 100 | 46 | | |
| **MAP4K5** | **MAP4K5** | **96** | **100** | **63** | **8** | **1257.42** | |
| MEK3 | MAP2K3 | 100 | 100 | 100 | 64 | | |
| **MEK4** | **MAP2K4** | **48** | **27** | **2.6** | **0.05** | **37.96** | |
| MEK6 | MAP2K6 | 82 | 100 | 100 | 47 | | |
| MINK | MINK1 | 89 | 100 | 98 | 55 | | |
| MKK7 | MAP2K7 | 100 | 100 | 100 | 84 | | |
| MLK1 | MAP3K9 | 100 | 100 | 100 | 100 | >10,000 | >5000 |
| MLK2 | MAP3K10 | 100 | 82 | 100 | 76 | | |
| MLK3 | MAP3K11 | 100 | 100 | 100 | 100 | | |
| MST1 | STK4 | 100 | 93 | 84 | 55 | 6709.79 | >5000 |
| OSR1 | OXSR1 | 100 | 94 | 95 | 42 | | |
| PAK1 | PAK1 | 93 | 97 | 83 | 22 | | |
| RIPK1 | RIPK1 | 99 | 87 | 85 | 50 | | |
| SRMS | SRMS | 1.9 | 0.55 | 0.05 | 0 | 0.64 | |
| STK39 | STK39 | 100 | 100 | 100 | 59 | | |
| TAK1 | MAP3K7 | 90 | 88 | 85 | 49 | | |
| TAOK1 | TAOK1 | 87 | 94 | 89 | 65 | 7532.57 | >5000 |
| TAOK2 | TAOK2 | 92 | 100 | 93 | 51 | | |
| TAOK3 | TAOK3 | 100 | 98 | 96 | 58 | | |
| TNIK | TNIK | 97 | 89 | 79 | 24 | | |
| **ZAK** | **ZAK** | **20** | **4** | **0.7** | **0.1** | **9.47** | |

Quantitative competitive binding assays were performed for a group of kinases previously tested against PLX4720 as well as a group of MAP kinases upstream of JNK. Published biochemical IC50s for PLX4720 are listed (see main text) for comparison and demonstrate good quantitative correspondence between estimated $K_d$ from binding assays and biochemical IC50s. ZAK and MKK4 (MAP2K4) were very tightly bound by PLX4720 with estimated $K_d$ below 50 nM. Bold text indicates the kinases tested for inhibition by PLX4720 with in-vitro kinase assays.

**Table 3.** Quantitative competitive binding assays reveal additional kinase targets of vemurafenib

| | | Percent control (50 nM) | Percent control (200 nM) | Percent control (1000 nM) | Percent control (10 μM) | Calculated estimate of IC50 (nM) | Published biochemical IC50 (nM) |
|---|---|---|---|---|---|---|---|
| ASK1 | MAP3K5 | 90 | 94 | 97 | 100 | 11,972.22 | >1000 |
| ASK2 | MAP3K6 | 94 | 98 | 100 | 74 | | |
| BLK | BLK | 96 | 66 | 30 | 0.55 | 518.03 | 547 |
| **BRAF(V600E)** | **BRAF** | **63** | **25** | **5.4** | **0.5** | **64.78** | **31** |
| BRK | PTK6 | 63 | 28 | 6.9 | 0.35 | 68.04 | 213 |
| DLK | MAP3K12 | 98 | 97 | 66 | 92 | | |
| FGR | FGR | 65 | 49 | 13 | 1.6 | 149.26 | 63 |
| HPK1 | MAP4K1 | 95 | 88 | 67 | 15 | | |
| LZK | MAP3K13 | 100 | 99 | 93 | 74 | | |
| MAP3K1 | MAP3K1 | 98 | 84 | 89 | 81 | | |
| MAP3K15 | MAP3K15 | 84 | 100 | 84 | 91 | | |
| MAP3K2 | MAP3K2 | 91 | 91 | 89 | 83 | | |
| MAP3K3 | MAP3K3 | 87 | 97 | 100 | 94 | | |
| MAP3K4 | MAP3K4 | 95 | 92 | 87 | 46 | | |
| MAP4K2 | MAP4K2 | 99 | 82 | 95 | 46 | | |
| MAP4K3 | MAP4K3 | 80 | 90 | 82 | 24 | | |
| MAP4K4 | MAP4K4 | 96 | 92 | 83 | 23 | 2842.34 | >1000 |
| **MAP4K5** | **MAP4K5** | **62** | **33** | **4.1** | **0.1** | **58.21** | **51** |
| MEK3 | MAP2K3 | 100 | 96 | 98 | 54 | | |
| **MEK4** | **MAP2K4** | **19** | **4.1** | **0.2** | **0.05** | **6.82** | |
| MEK6 | MAP2K6 | 91 | 97 | 87 | 21 | 4080.69 | >1000 |
| MINK | MINK1 | 100 | 100 | 91 | 66 | 14,761.44 | >1000 |
| MKK7 | MAP2K7 | 97 | 95 | 94 | 85 | | |
| MLK1 | MAP3K9 | 100 | 93 | 97 | 41 | 13,979.88 | >1000 |
| MLK2 | MAP3K10 | 92 | 96 | 87 | 78 | | |
| MLK3 | MAP3K11 | 98 | 100 | 100 | 77 | | |
| MST1 | STK4 | 99 | 83 | 51 | 12 | | |
| OSR1 | OXSR1 | 100 | 100 | 89 | 98 | | |
| PAK1 | PAK1 | 99 | 98 | 91 | 46 | | |
| RIPK1 | RIPK1 | 92 | 100 | 99 | 73 | | |
| SRMS | SRMS | 24 | 9.6 | 0.75 | 0 | 11.15 | 18 |
| STK39 | STK39 | 100 | 100 | 97 | 66 | | |
| TAK1 | MAP3K7 | 93 | 88 | 86 | 88 | | |
| TAOK1 | TAOK1 | 91 | 100 | 97 | 79 | | |
| TAOK2 | TAOK2 | 98 | 92 | 95 | 70 | 11,770.83 | >1000 |
| TAOK3 | TAOK3 | 92 | 98 | 92 | 80 | 15,468.75 | >1000 |
| TNIK | TNIK | 95 | 94 | 66 | 11 | | |
| **ZAK** | **ZAK** | **9** | **1.8** | **0.25** | **0.05** | **4.03** | |

Quantitative competitive binding assays were performed for a group of kinases previously tested against vemurafenib as well as a group of MAP kinases upstream of JNK. Published biochemical IC50s for vemurafenib are listed (see main text) for comparison and demonstrate good quantitative correspondence between estimated $K_d$ from binding assays and biochemical IC50s. ZAK and MKK4 (MAP2K4) were very tightly bound by vemurafenib with estimated $K_d$ below 50 nM. Bold text indicates the kinases tested for inhibition by vemurafenib with in-vitro kinase assays.

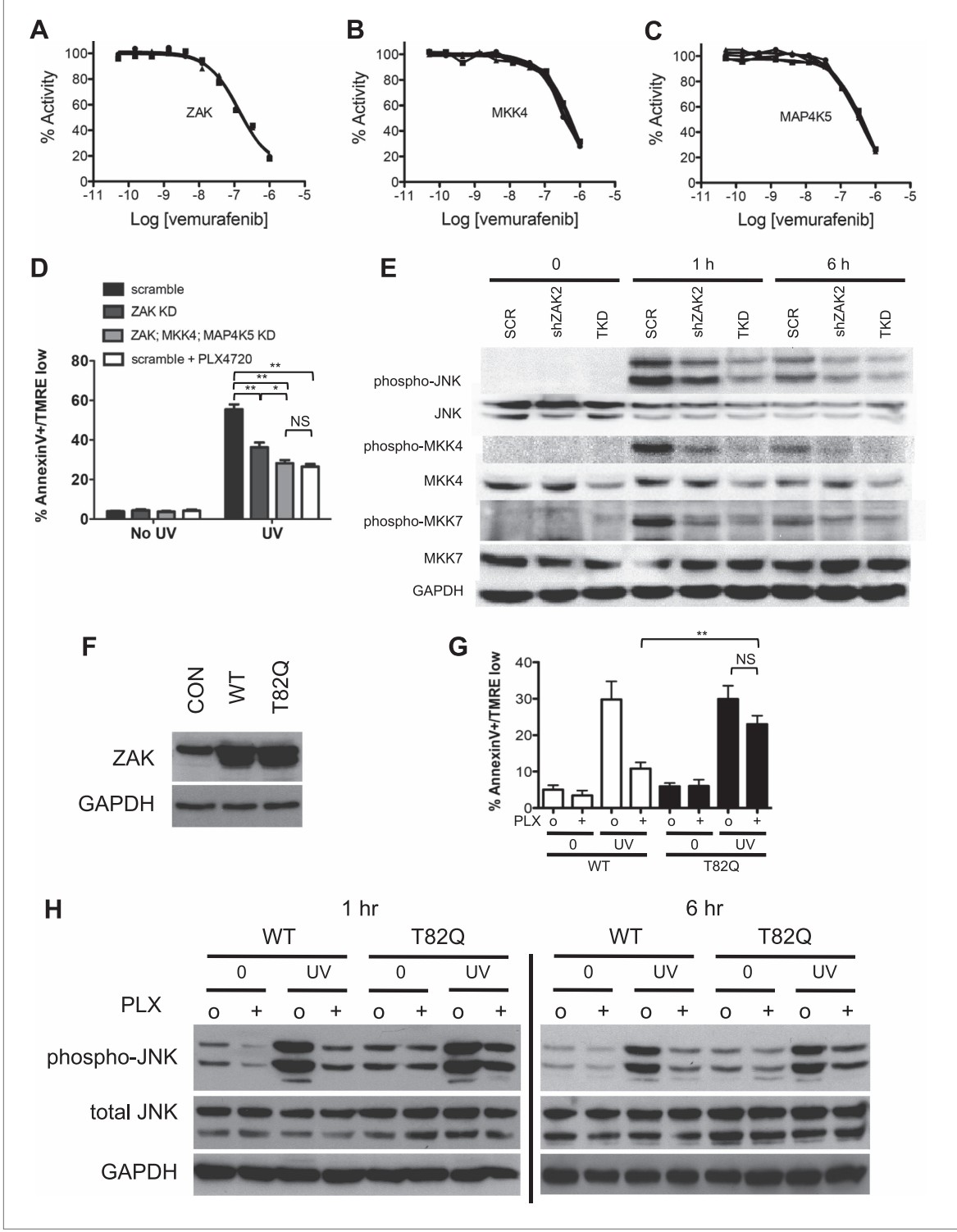

**Figure 3**. PLX4720 and vemurafenib suppress apoptosis and JNK signaling through inhibition of off-target kinases. (**A–C**) In-vitro kinase assays for ZAK, MKK4, and MAP4K5 were performed across a 10-point concentration range from 0.05 to 1000 nM in triplicate, revealing significant inhibition of kinase activity within the nM range for vemurafenib. (**D**) Lentiviral shRNA knockdown of ZAK singly or in combination with MKK4 and MAP4K5 (triple knockdown, 'TKD') was performed revealing potent suppression of apoptosis as measured by FACS for Annexin V+, TMRE-low cells (n = 5, '*' denotes statistical significance at p<0.05, '**' at p<0.01, 'NS' is not significant) at 24 hr following single dose UVB irradiation at 720 J/m². ZAK knockdown and triple knockdown cells exhibit 70% and 94% suppression of apoptosis, respectively, relative to PLX4720-treated cells expressing a non-suppressing shRNA control (scramble, 'SCR'). (**E**) Western blots of lysates obtained at 1 and 6 hr post-UV irradiation show potent induction of

*Figure 3. Continued on next page*

*Figure 3. Continued*

phospho-MKK4, phospho-MKK7, and phospho-JNK which are all suppressed with progressively increasing effect in ZAK single knockdown ('shZAK2') and triple knockdown ('TKD') HaCaT cells. (**F**) Western blots of HaCaT cells electroporated with pcDNA3-wild-type (WT) ZAK and the gatekeeper mutant pcDNA3-(T82Q) ZAK show equivalent expression. (**G**) HaCaT cells overexpressing ZAK (WT) and ZAK (T82Q) were irradiated with a single dose of UVB irradiation at 720 J/m$^2$ in the absence ('o') and presence ('+') of 1 μM PLX4720 and apoptosis measured by FACS for Annexin V+, TMRE-low cells (n = 4, '**' at p<0.01, 'NS' is not significant) at 24 hr. ZAK (WT) cells are sensitive to PLX4720-mediated suppression of apoptosis (bar 3 vs 4), but drug-treated ZAK (T82Q)-expressing cells undergo significantly more apoptosis than drug-treated ZAK (WT) cells (bar 4 vs 8), with bypass of PLX4720-induced suppression as compared to drug-treated ZAK (WT) cells (paired t-test, p=0.005). (**H**) Western blots of ZAK (WT) and ZAK (T82Q)-expressing HaCaT cells at 1 hr and 6 hr post-irradiation show that phospho-JNK activation is intact in both cell lines in the absence of drug (lanes 3, 7), but that drug-treated ZAK (T82Q)-expressing HaCaT cells have significantly more phospho-JNK activation at both 1 and 6 hr post-irradiation, as compared to drug-treated ZAK (WT)-expressing cells (lane 4 vs 8).

The following figure supplements are available for figure 3:

**Figure supplement 1**. Knockdown of ZAK potently inhibits JNK activation and UV-induced apoptosis.

**Figure supplement 2**. Single knockdown of MKK4 or MAP4K5 partially inhibits JNK activation and UV-induced apoptosis.

**Figure supplement 3**. Knockdown of ZAK potently inhibits JNK activation and UV-induced apoptosis in SRB1 cells.

**Figure supplement 4**. Vemurafenib and PLX4720 inhibit multiple kinases upstream of JNK and p38.

## Vemurafenib suppresses JNK activity and apoptosis in cSCC arising in treated patients

We then explored whether vemurafenib or PLX4720-mediated suppression of JNK and apoptosis is relevant in vivo. We first examined cSCC arising in patients treated with vemurafenib and compared them to sporadic cSCC that were histologically similar, arising in individuals never treated with vemurafenib (*Figure 4A–E*). Phospho-JNK and cleaved caspase-3 expression were assessed by immunohistochemistry and then quantified following normalization by unit area (mm$^2$) of tumor tissue (malignant keratinocytes) only (*Figure 4A–D*, *Figure 4—figure supplement 1*). Sporadic cSCC arising in patients never treated with vemurafenib (n = 15) contained substantially greater expression of phospho-JNK (p=0.013; *Figure 4A,E*) and cleaved caspase-3 (p=0.042; *Figure 4C,E*) as compared to lesions arising in vemurafenib-treated patients (n = 16; *Figure 4B,D,E*). Therefore, we found significant reductions in phospho-JNK and cleaved caspase-3 expression in human cSCC suggesting that suppression of JNK activity and apoptosis occur in vivo in patients treated with vemurafenib.

## BRAFi suppress acute UVR-induced epidermal apoptosis in vivo

We then probed the acute, in vivo, short-term UV response in skin by pre-treating C57BL/6 mice with PLX4720 administered by oral gavage 40–80 mg/kg twice a day for 2–4 days (*Tsai et al., 2008*). Following depilation, mice were irradiated once using a solar simulator (Oriel) with 10 kJ/m$^2$ UVB. The skin was harvested at 1 hr, 6 hr, and 24 hr post-irradiation. Consistent with our other results, we found significant apoptosis of epidermal keratinocytes in irradiated mouse skin that was suppressed by PLX4720 treatment (12.7 ± 0.4 apoptotic cells/mm vs 4.9 ± 0.3 apoptotic cells/mm skin, [n = 3 pairs], p<10$^{-5}$; *Figure 4F,G*), a finding corroborated by cleaved caspase-3 levels, which were induced within 6 hr of irradiation and suppressed in PLX4720-treated mice (*Figure 4H*). As expected, phospho-JNK and phospho-p38 were significantly upregulated following UV irradiation and phospho-JNK was significantly suppressed by PLX4720 (*Figure 4H*). The upstream kinases MKK4 and MKK7 were likewise activated by UV radiation and suppressed by PLX4720 at 1 and 6 hr post-irradiation, confirming the importance of this mechanism of PLX4720-induced JNK signaling suppression in vivo. Finally, Noxa mRNA expression, as measured by qPCR, was strongly induced by UV exposure, a response significantly dampened by PLX4720 treatment (*Figure 4I*).

## BRAFi accelerates UVR-driven cSCC development in Hairless mice

We also used the Hairless mouse model of squamous cell carcinoma to assess whether PLX4720 would affect UV-driven tumor development. This is particularly relevant since it appears that UV exposure is

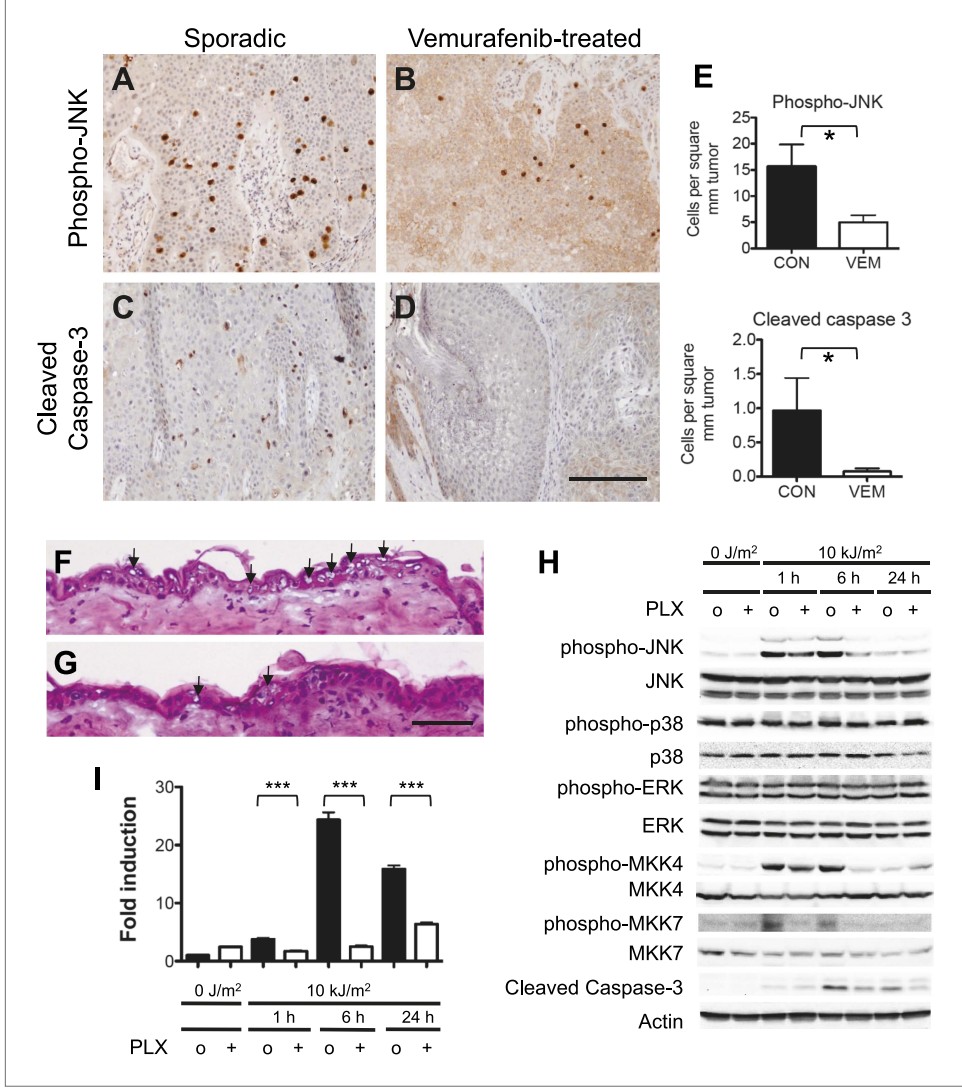

**Figure 4**. Vemurafenib and PLX4720 suppress apoptosis and JNK signaling in vivo. (**A–D**) cSCC samples from vemurafenib-treated patients and non-treated patients were analyzed by immunohistochemistry for phospho-JNK and cleaved caspase-3 expression. cSCC arising in vemurafenib-treated patients show decreased expression of phospho-JNK (**B**) and cleaved caspase-3 (**D**) as compared to sporadic cSCC in patient never treated with vemurafenib (**A** and **C**). Scale bar is 100 μm. (**E**) Comparisons of stained cells normalized to mm$^2$ of tumor area revealed significant suppression of both phospho-JNK and cleaved caspase 3 expression in vemurafenib-treated cSCC ('*', p<0.05). (**F** and **G**) Hematoxylin-stained cryosections of skin harvested at 24 hr post-irradiation showed extensive apoptosis (arrowheads) with vacuolated blebbed cells and clumped pyknotic nuclei in control-treated mice (**F**) and significantly fewer apoptotic cells in PLX4720-treated mice (**G**). Scale bar is 50 μm. (**H–I**) Vehicle-treated ('o') and PLX4720-treated ('+') mice were unirradiated or irradiated once, and epidermis was harvested at 1 hr, 6 hr, and 24 hr post-irradiation. (**H**) Significant UV-induced upregulation of both phospho-JNK and phospho-p38 were observed within 1 hr, with significant suppression of phospho-JNK in PLX4720-treated mice by 6 hr and minimal suppression of phospho-p38. Phospho-ERK levels remained constant. The upstream regulators of JNK, MKK4 and MKK7, were both significantly activated within 1 hr of irradiation, and potently suppressed in PLX4720-treated mice. Cleaved caspase-3 levels increased within 6 hr and were suppressed in PLX4720-treated mice. (**I**) Noxa was induced most significantly at 6 hr and was potently suppressed by PLX4720 at all time points ('***', p<0.001).

The following figure supplements are available for figure 4:

**Figure supplement 1**. Double staining of sporadic cSCC confirms phospho-JNK and cleaved caspase-3 expression within keratinocytes of tumors.

an important initiating event in BRAFi-accelerated cSCC (*Su et al., 2012*). Unlike the DMBA/TPA model, in which lesions almost universally harbor *Hras* mutations (*Brown et al., 1990*), the Hairless model has a very low frequency of *Ras* mutation in papillomas and carcinomas (*van Kranen et al., 1995*), more similar to sporadic human cSCC. The cohorts (n = 5 each) were identically irradiated thrice weekly (12.5 kJ/m$^2$ per week UVB) for 72 days before starting on PLX4720 treatment vs vehicle control. Within 20 days of administration of drug, hyperkeratotic papules were visible on the backs of PLX4720-treated animals (*Figure 5A,B*), which steadily grew into cSCC over the following several weeks (*Figure 5C,D*). Within this period of 150 days (78 days of drug treatment), control-treated mice had not yet developed any visible lesions (*Figure 5E*). When we quantified the effects of each of these drug treatments, we found significant decreases in both phospho-JNK expression (p=0.046; *Figure 5F,G,J*) and cleaved caspase 3 expression (p=0.019; *Figure 5H–J*) in PLX4720-treated mice as compared to control-treated mice. Importantly, we sequenced the entire coding regions for *Ras* (*Hras*, *Kras*, *Nras*) and found no mutations in any of the tumors in PLX4720-treated mice, as compared to one of 14 papillomas and carcinomas in a cohort of control-treated chronically-irradiated Hairless mice (*Figure 5—figure supplement 1*).

## Paradoxical ERK activation and off-target JNK inhibition cooperate to accelerate tumor growth

While the effects of these BRAFi on JNK-dependent apoptosis is clear and independent of ERK activity, the relative contribution of paradoxical ERK activation vs JNK pathway inhibition to tumorigenesis has not been precisely addressed (*Figure 6A*). To accomplish this, we took advantage of the fact that paradoxical ERK activation requires intact *CRAF* (*Hatzivassiliou et al., 2010*; *Heidorn et al., 2010*; *Poulikakos et al., 2010*) (*Figure 6A*). We used isogenic, matched WT and *Craf*-deficient (*Craf–/–*) mouse embryonic fibroblasts (MEFs) and transformed them with adenovirus E1A and human *HRAS*$^{G12V}$ to enable anchorage-independent growth (*Figure 6—figure supplement 1*). These *Craf–/–* cells do not exhibit strong paradoxical MEK or ERK activation, consistent with previous reports (*Poulikakos et al., 2010*) (*Figure 6—figure supplement 1A*). Wild-type and matched *Craf*-deficient MEFs were plated in soft agar assays (*Su et al., 2012*) and treated with PLX4720. Both WT and *Craf*-deficient MEFs exhibited a significant colony formation advantage in the presence of drug (*Figure 6B–D*). Based upon this analysis, we estimated that the effect of paradoxical ERK activation to be 60% and other effects, including inhibition of JNK activity, to account for the rest (40%) of the total colony growth advantage (*Figure 6D*). To assess the role of JNK signaling directly, we used *HRAS*$^{G12V}$-transformed HaCaT cells with ('TKD') and without ('SCR') triple lentiviral knockdown of ZAK, MAP4K5, and MKK4 (*Figure 3D,E*), to perform similar colony formation assays to assess responses to PLX4720 treatment (*Figure 6E*). Drug treatment conferred a significant colony formation advantage in both sets of cells, which exhibit equivalent paradoxical ERK activation (*Figure 6—figure supplement 1B*). Yet, untreated TKD HaCaT cells produced more colonies than SCR HaCaT cells suggesting that JNK pathway suppression results in an advantage in the absence of drug and paradoxical ERK activation (*Figure 6E*). Drug-treated SCR and TKD HaCaT cells, had elevated colony counts to similar levels, as expected, because both lines would experience similar degrees of both paradoxical ERK activation and JNK inhibition, and TKD cells (knocked down for ZAK, MAP4K5, MKK4) are unlikely to experience any further suppression of JNK signaling (*Figure 3D*). Based on this, we estimated the effect of JNK pathway inhibition to be 17.6% (*Figure 6E*). When combined with the MEF experiment, we estimate that the effect of JNK inhibition contributes approximately 17.6–40% of the total effect of PLX4720-accelerated colony formation (*Figure 6D,E*). Importantly, although we can quantify these individual contributions, it is clear in many contexts in cancer that hyperproliferation and inhibition of apoptosis are highly cooperative (*Hanahan and Weinberg, 2011*), and our data do not preclude the possibility that one or both are individually required.

## Dabrafenib and vemurafenib differ significantly in their off-target effects and risk of cSCC

The recently updated combination trial of the BRAFi dabrafenib and MEKi trametinib shows a low 7% cSCC rate in 54 patients, (*Flaherty et al., 2012*) suggesting that combined MEK inhibition can reduce, but not eliminate cSCC formation, nevertheless reinforcing a role for paradoxical ERK activation (*Su et al., 2012*). However, clinical trial data on dabrafenib alone at 150 mg PO BID, shows an overall aggregated cSCC rate of 6.1% (*Falchook et al., 2012*; *Hauschild et al., 2012*; *Long et al., 2012*) vs 22%

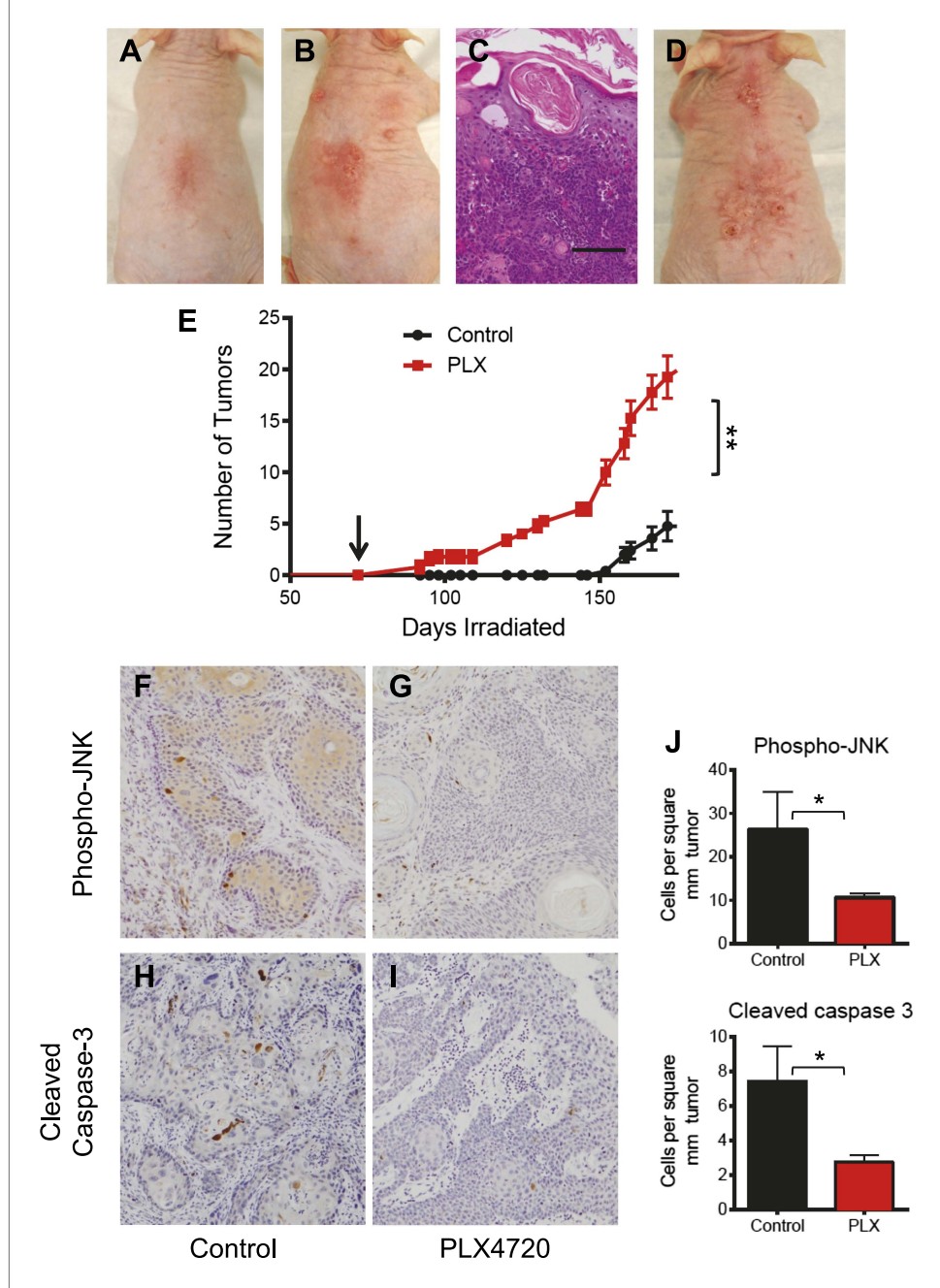

**Figure 5**. PLX4720 and JNK inhibition dramatically accelerate cSCC development in the UV-driven Hairless mouse model. (**A–E**) Chronically-irradiated Hairless mice were treated with PLX4720 (n = 5), or vehicle (n = 5) starting at day 72 (arrow, **E**). Tumors were induced within 20 days of PLX-4720 treatment (**B**), whereas only erythema was seen in control animals (**A**). The tumors in PLX4720-treated mice progressed to well-differentiated cSCC (**C**, scale bar 75 µm), steadily increasing in size and number (**D**, day 132). (**E**) Even at 150 days (78 days of drug treatment), only PLX4720-treated mice had tumors and the differences in tumor number persisted throughout ('**', p=0.0026). (**F–J**) cSCC from mice were harvested and assessed for phospho-JNK and cleaved caspase 3 expression by immunohistochemistry. Tumors from PLX4720-treated animals showed significantly lower levels of phospho-JNK (**G**) and cleaved caspase 3 (**I**) as compared to control-treated animals (**F** and **H**). Differences in these parameters were significant across all comparisons (**J**, '*', p<0.05).

The following figure supplements are available for figure 5:

**Figure supplement 1**. cSCC and papillomas arising in Hairless mice treated with PLX4720 do not have *Ras* mutations.

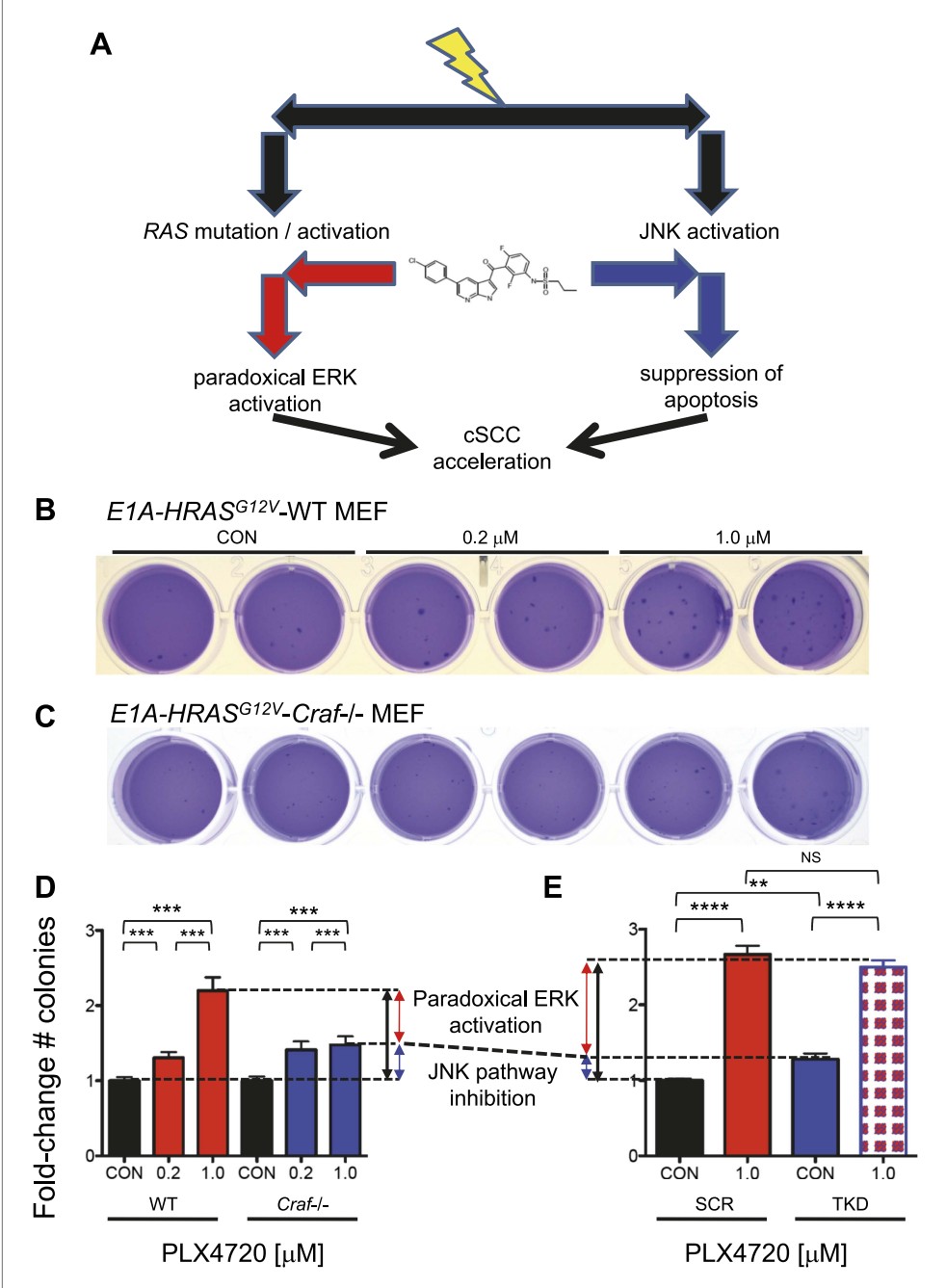

**Figure 6**. Paradoxical ERK activation and JNK pathway inhibition make significant and separable contributions to BRAFi-induced growth. (**A**) We envision two separable, parallel mechanisms by which PLX4720 and vemurafenib contribute to cSCC development. Drug-induced paradoxical ERK activation and inhibition of JNK signaling occur in parallel, but the former depends on intact *CRAF*. (**B** and **C**) Representative soft agar colonies of E1A and *HRAS*$^{G12V}$-transformed wild-type (WT) (**B**) and *Craf−/−* (**C**) MEFs, following exposure to 0.2 μM and 1.0 μM PLX4720 over 4–6 weeks show significant colony-forming advantages conferred by BRAFi. (**D**) The fold-change in colony counts of transformed wild-type (WT) (n = 22 replicates) and *Craf−/−* (n = 14 replicates) MEFs demonstrate a dose-dependent increase in colonies, particularly for WT MEFs. The difference between colony formation advantages conferred by 1.0 μM PLX4720 in WT vs *Craf−/−* MEFs was interpreted to reflect the contribution of paradoxical ERK signaling (red arrow), which depends upon *Craf*, and is 60% of the total effect (black arrow), with the remainder composed of other effects including JNK inhibition (blue arrow). All differences between each MEF population were significant ('***', p<0.001) (**E**) The fold-change in colony counts of transformed HaCaT cells with

*Figure 6. Continued on next page*

*Figure 6. Continued*

('TKD') and without ('SCR') triple lentiviral shRNA knockdown of ZAK, MAP4K5, and MAP2K4, show significant differences between 1.0 µM PLX4720-treated and control-treated conditions ('****', $p<10^{-10}$). Importantly, untreated TKD cells had a significant advantage over untreated SCR HaCaT cells ('**', $p<0.01$), which we interpreted to be the contribution of JNK signaling inhibition, of 17.6% (blue arrow). Drug-treated SCR and TKD cells both had a similar degree of total colony formation advantage (averaged as black arrow), as expected, since the TKD cells are not expected to have any additional suppression of JNK signaling in the presence of drug ('NS', $p=0.17$, *Figure 3D*). Therefore, the colony counts for these two distinct systems (**D** and **E**), when taken together, show that JNK pathway inhibition accounts for approximately 17.6–40% and paradoxical ERK activation accounts for approximately 60–82.4% of the total effects of PLX4720 on tumor growth.

The following figure supplements are available for figure 6:

**Figure supplement 1**. Paradoxical MEK and ERK activation require intact *Craf*.

**Figure supplement 2**. Dabrafenib fails to suppress apoptosis and phospho-JNK upregulation following UV irradiation at bioequivalent doses as compared to PLX4720.

**Figure supplement 3**. Dabrafenib produces a colony formation advantage only in WT MEFs.

for vemurafenib at 960 mg PO BID in several hundred patients (*Flaherty et al., 2010*; *Chapman et al., 2011*; *Sosman et al., 2012*; *Menzies et al., 2013*). We interpreted this as being reflective of differences between vemurafenib and dabrafenib, as opposed to unequivocal proof that paradoxical ERK activation is the only mechanism involved. To explore this further, we used similar assays to assess the effects of dabrafenib on apoptosis, JNK signaling, and colony formation. In stark contrast to vemurafenib, dabrafenib has little effect on apoptosis and JNK signaling at doses that are biologically equivalent based upon growth inhibition of $BRAF^{V600E}$ melanoma cells and human pharmacokinetic data (*Falchook et al., 2012*; *Gowrishankar et al., 2012*) (*Figure 6—figure supplement 2A*). Peak serum concentrations of dabrafenib at 150 mg PO BID in humans (*Falchook et al., 2012*) are over 50-fold lower (1.55 µM) than mean sustained serum levels of vemurafenib (86 µM) at 960 mg PO BID (*Flaherty et al., 2010*), and the $GI_{50}$ for the A375 melanoma cell line is less than 0.01 µM for dabrafenib (*Greger et al., 2012*) vs 0.50 µM for PLX4720 (*Tsai et al., 2008*). Even at 0.05 µM, dabrafenib did not significantly impact UV-induced phospho-JNK upregulation or apoptosis in HaCaT and SRB1 cells (*Figure 6—figure supplement 2B–E*). We profiled dabrafenib activity against ZAK, MKK4, and MAP4K5, and found that ZAK is a significant off-target kinase for dabrafenib as well, but at 0.01 µM, over 64% of activity is retained (*Figure 6—figure supplement 2F*). Neither MKK4 nor MAP4K5 is substantially inhibited by dabrafenib up to 1 µM (*Figure 6—figure supplement 2F*).

Using transformed WT and *Craf*-deficient MEFs in soft agar assays, we also showed that dabrafenib enhanced colony formation in WT MEFs, but not in *Craf*-deficient MEFs (*Figure 6—figure supplement 3*). Our results suggest that while both dabrafenib and vemurafenib cause equivalent paradoxical ERK activation in *BRAF*-wild-type cells (*Figure 6—figure supplement 1A–B*), only vemurafenib confers a significant colony formation advantage in *Craf*-deficient cells that have no significant paradoxical MEK/ERK activation, implicating off-target effects as a key difference between the two drugs with respect to cSCC development (*Menzies et al., 2013*).

## Discussion

We have discovered an unexpected and novel effect of the BRAFi PLX4720 and vemurafenib in inhibiting apoptosis in vitro and in vivo through the ERK-independent suppression of JNK signaling (*Tournier et al., 2000*). Our studies implicate the off-target binding and inhibition of these compounds to ZAK primarily (*Figure 3—figure supplements 1–3*), with additional contributions of MKK4 (MAP2K4) and MAP4K5 inhibition, thus implicating inhibition of JNK signaling at all three upstream tiers of MAP kinase signaling (*Figure 3—figure supplement 4*). Although MKK4 knockdown alone could suppress UV-induced apoptosis and phospho-JNK induction by up to 27.3% (*Figure 3—figure supplement 2*), this is expected, given that ZAK signals through MKK4 and MKK7 (*Gross et al., 2002*) (*Figure 2H,I,3E,4H*) and MKK4 is important (with MKK7) for full JNK activation (*Tournier et al., 2001*; *Haeusgen et al., 2011*). Additionally, UV-mediated induction of NOXA is suppressed in cell lines, primary NHEKs, and

in vivo, indicating that this BCL2 family member may be a critical effector of apoptosis in this context (*Figures 1G and 4I*). In chronically-irradiated Hairless mice, development of well-differentiated papillomas and cSCC is substantially accelerated by PLX4720 treatment without the need for *Ras* mutation and with a dramatic reduction in latency by at least 10 weeks (*Figure 5E*, *Figure 5—figure supplement 1*).

While there is enrichment for *RAS* mutations in human cSCC arising in vemurafenib-treated patients vs controls (*Oberholzer et al., 2011*; *Su et al., 2012*), up to 30–40% of these lesions do not have *RAS* mutations. Our novel mechanism of BRAFi-mediated apoptosis suppression is the off-target inhibition of several kinases in the JNK pathway, which is independent from, and compatible with, paradoxical ERK–dependent mechanisms (*Su et al., 2012*) (*Figure 6A*). Importantly, our approach (*Figure 6B–E*) has not only allowed us to quantify the contribution of the effect on apoptosis (17.6–40%) vs paradoxical ERK activation (60–82.4%), but also shows that the growth advantage conferred by BRAFi in *BRAF*-WT cells is not accounted for entirely by paradoxical ERK activation. Our results have also shown that there are significant differences between the BRAFi vemurafenib and dabrafenib (*Figure 6—figure supplements 2,3*) with respect to these off-target effects in cells (even though their relative selectivities for BRAF over ZAK are similar) and this may, in part, explain why they differ in rates of cSCC (*Flaherty et al., 2010*; *Chapman et al., 2011*; *Falchook et al., 2012*; *Hauschild et al., 2012*; *Long et al., 2012*; *Sosman et al., 2012*). At present it is unclear why ZAK appears to be a common off-target kinase and whether structural similarities with other kinases may explain this (*Sauter et al., 2010*; *Wong et al., 2013*).

ZAK has been previously studied in the context of bacterial toxin and doxorubicin-mediated cytokine signaling (*Jandhyala et al., 2008*; *Sauter et al., 2010*; *Stone et al., 2012*; *Wong et al., 2013*), cardiac (*Huang et al., 2004*) and ischemic stress responses (*Su et al., 2012*), and in cellular responses to ionizing radiation (*Gross et al., 2002*; *Tosti et al., 2004*; *Vanan et al., 2012*). It is widely expressed across tissues including epidermis, but most prominently in heart, liver, and muscle (*Abe et al., 1995*; *Miyata et al., 1999*; *Liu et al., 2000*; *Bloem et al., 2001*; *Gross et al., 2002*; *Su et al., 2004*), and has purported tumor suppressive roles in lung cancer (*Yang et al., 2010*) and tumor promoting ones in partially transformed mouse skin epidermal cells (*Cho et al., 2004*). ZAK is a MAP3K that is upstream of both JNK and p38 signaling (*Gross et al., 2002*; *Tosti et al., 2004*; *Jandhyala et al., 2008*; *Cheng et al., 2009*; *Stone et al., 2012*; *Wong et al., 2013*) and signals to JNK through MKK4 and MKK7 (*Gross et al., 2002*) (*Figures 2H,I,3E,4H*). Accordingly, macrophages derived from ZAK-deficient mice have profound defects in activation of both JNK and p38 signaling following doxorubicin exposure (*Wong et al., 2013*). In the setting of UV-induced apoptosis as we have examined here, JNK activity is the major driver of apoptosis (*Derijard et al., 1994*; *Chen et al., 1996*; *Tournier et al., 2000*), also by virtue of the fact that phospho-p38 induction by UV is inconstant (*Figures 1F,H,2B,4H*); although where it is induced, PLX4720/vemurafenib treatment suppresses it (*Figure 1F* (SRB1, HaCaT cells), *Figures 1H,2B*). These results are consistent with the model (*Figure 3—figure supplement 4*) that ZAK signals to both JNK and p38, but is principally necessary for activating JNK in stress-induced apoptosis.

Because off-target kinases in the JNK pathway are affected by vemurafenib/PLX4720, one expects that these kinases would be affected in all cells regardless of *BRAF* status. Indeed, melanoma cells expressing BRAF[V600E] also exhibit suppression of JNK activity following irradiation (*Figure 1H*). However, in *BRAF*[V600E]-expressing melanoma cells, the effect of blocking BRAF activity alone clearly dominates, because these cells are exquisitely dependent upon BRAF activity (*Tsai et al., 2008*). Therefore, although off-target kinases are inhibited, the cellular context of dependence on particular kinases is still highly relevant and likely dictates the outcome.

Our findings suggest a tumor suppressive role for JNK signaling in the context of drug-induced cSCC, though the role of JNK in cancer is highly context-dependent and is partly related to differing functions of the individual isoforms and partial redundancy (*Tournier et al., 2000*). Nonetheless, there is ample in vivo evidence showing that JNK can function in a tumor suppressive role. Genetically-engineered mice lacking *Jnk1* and *Jnk2* have increased (*She et al., 2002*) and decreased (*Chen et al., 2001*) susceptibility, respectively, to chemical carcinogenesis in skin, though these mice also have opposite defects in epidermal differentiation (*Weston et al., 2004*). In mouse models, lack of *Jnk1/2* activity suppresses *Ras*-driven tumorigenesis in lung (*Cellurale et al., 2011*) and promotes it in *Ras*-driven and *Trp53*-deficient breast cancer models (*Cellurale et al., 2010, 2012*). In the context of *Pten*-deficiency, loss of *Jnk1/2* or *Mkk4/Mkk7* promotes aggressive prostate adenocarcinoma (*Hubner et al., 2012*).

Importantly, the effects of JNK on cancer are not always tumor cell autonomous, as JNK activity supports a pro-tumorigenic inflammatory microenvironment in hepatocellular carcinoma (*Das et al., 2011*).

Our results have important clinical implications and suggest careful consideration of combining certain BRAFi with therapeutic modalities that induce apoptosis such as radiation or chemotherapy, particularly with respect to off-target tissues (keratinocytes in skin). We have shown that off-target inhibition of kinases, even at higher IC50s, can contribute biologically significant effects, particularly if they are in the same pathway. Finally, our results show that kinase inhibitors must be considered in terms of their entire spectrum of activity, which can dramatically affect pathways distinct from those affected by inhibition of the intended target.

## Materials and methods

### Ethics statement
All studies were conducted under institutionally-approved IRB (LAB08-0750) and ACUF (06-09-06332) protocols for the protection of human and animal subjects, respectively.

### Cell lines
Cutaneous SCC cell lines (SRB1, SRB12, COLO16) were obtained from Jeffrey N Myers (MD Anderson), HaCaT cells from Norbert Fusenig (German Cancer Research Center), and WM35 and A375 melanoma cell lines from Michael Davies (MD Anderson). The cell lines were validated by STR DNA fingerprinting using the AmpFlSTR Identifiler kit according to manufacturer instructions (Applied Biosystems, Grand Island, NY). The STR profiles were compared to known ATCC fingerprints (ATCC.org), to the Cell Line Integrated Molecular Authentication database (CLIMA) version 0.1.200808 (http://bioinformatics.istge.it/clima/) and to the MD Anderson fingerprint database. The STR profiles matched known DNA fingerprints (HaCaT) or were unique (SRB1, SRB12, COLO16). The cells were cultured in DMEM/Ham's F12 50/50 (Cellgro) supplemented with 10% Fetal Bovine Serum (FBS) (Sigma), glutamine, and Primocin (Invivogen). NHEKs (Lonza) were cultured in media according to manufacturer's instructions. Irradiation was performed using an FS40 sunlamp dosed by an IL1700 radiometer. Following irradiation, cells were treated with PLX4720 (Plexxikon), vemurafenib (Selleck Chemicals) or DMSO (1:2000).

### Antibodies
Primary antibodies (Cell Signaling) used for Western blot analysis included p53 (2527P, clone 7F5), phospho-/total p44/42 MAPK (4370S, cloneD13.14.4E/9102S), phospho-/total p38 MAPK (4511S, clone D3F9/9212S), phospho-/total JNK (4668S, clone 81E11/9252S), BIM (2933, clone C34C5), MCL1 (5453P, clone D35A5), cleaved caspase-3 (9661L, clone D175), phospho-/total MKK7 (4171S/4172S), phospho-/total MKK4 (9156S/9152S), phospho-/total MEK (9121S/9122), MAP4K5 (ab56848; Abcam) and NOXA (mA1-41000; Thermo Scientific). GAPDH (21,182, clone 14C10; Cell Signaling) and beta-actin (A5060; Sigma) were probed to ensure even loading of protein samples. Immunohistochemistry was performed for phospho-JNK (V7931; Promega) and cleaved caspase-3 (Cell Signaling as above). Antibody against ZAK was generously provided by R Ruggieri (Feinstein Institute for Medical Research).

### Flow Cytometry
TMRE (Invitrogen) was used as a measure of mitochondrial membrane potential, Annexin V-FITC or Annexin V-APC (Invitrogen) as a probe for apoptosis, and Sytox Blue (Invitrogen) as an indicator for dead cells. At 24 hr post-irradiation, floating and adherent cells were collected and stained with TMRE, Annexin V and Sytox Blue. Data was collected and analyzed using a flow cytometer (Fortessa, Becton Dickinson) and FlowJo Software (Tree Star). Data were calculated and charts were plotted using GraphPad Prism 5 software.

### Western blot analysis
Cell were lysed in standard buffers with protease inhibitors (Roche) and phosphatase inhibitors (Santa Cruz) with extracts run on SDS/polyacrylamide gels and transferred to Immobilon-P transfer membrane (Millipore). Blots were blocked in TBST (10 mM Tris-HCL pH8, 150 mM NaCl, 0.5% Tween) with milk or BSA, probed with primary antibodies, corresponding HRP-conjugated secondary antibodies, and signals detected using ECL kit (Amersham).

## Immunohistochemistry and histology

Cutaneous squamous cell carcinomas biopsied from patients treated with or without BRAF-inhibitor were obtained either under clinical trials (Roche) or separate IRB approval (LAB08-0750). Staining levels were quantified by counting positively labeled cells and dividing by the total area of the tumor tissue within each sample. To measure tumor areas, all samples were photographed, tumor cells outlined, and total pixel numbers calculated using included image analysis tools in Adobe Photoshop and standardized to a hemacytometer to convert to $mm^2$. To measure apoptosis in irradiated skin, pyknotic or dyskeratotic epidermal keratinocytes were counted and normalized to length (mm) of epidermis.

## Kinase activity profiling

PLX4720 and vemurafenib were prepared in DMSO and tested in duplicate at four concentrations (50 nM, 200 nM, 1000 nM, 10 µM) against a panel of 38 kinases using a quantitative competitive binding assay (KINOMEscan, San Diego, CA). Average percent inhibition was reported. Estimated $K_d$ values were derived by averaging pointwise estimates calculated using a transformed Hill equation at each concentration of drug. In vitro kinase assays were performed using human full-length ZAK (MBP substrate, ATP 2.5 µM), amino acids 33-end MKK4 (JNK1 substrate, ATP 0.1 µM), and full length MAP4K5 (MBP substrate, ATP 10 µM) (Reaction Biology). Assays for BRAF$^{V600E}$ and ASK1 against vemurafenib were run in parallel revealing IC50s of 31.6 ± 2.9 nM for BRAF$^{V600E}$ and no significant inhibition of ASK1, as previously reported (*Bollag et al., 2010*).

## Lentiviral knockdown experiments

Lentiviral shRNA knockdown was accomplished using standard lentiviral methods using 293T cells and psPAX2/pVSV.G packaging plasmids. shRNA clones against ZAK (clones V2LHS_239842, V3LHS_336769), MKK4 (clones V3LHS_646205, V3LHS_386825A), and MAP4K5 (clones 196277A, 334084), as well as a non-silencing shRNA were obtained from Open Biosystems in the GIPZ vector. Following transduction, cells were puromycin-selected and FACS sorted to obtain cells with high-level suppression. Degree of mRNA suppression was quantified by qPCR using Taqman probes using internally controlled (2-color, same well) GAPDH probes to ensure proper normalization.

## ZAK overexpression

ZAK (T82Q) mutant was generated in the pcDNA3 mammalian expression vector. HaCaT cells were electroporated using the Neon transfection system 24 hr prior to irradiation. Transfection efficiencies were estimated to be 70–80% by GFP fluorescence.

## Mouse experiments

Wild-type C57BL/6 mice, 5–8 weeks old, were pretreated with PLX4720 in 5% DMSO in 1% methylcellulose for 2–4 days at 40–80 mg/kg twice a day or control 5% DMSO in 1% methylcellulose by oral gavage. The mice were shaved and depilated (Nair) 24 hr prior to irradiation with a solar simulator (Oriel) dosed at 10 kJ/$m^2$ of UVB. Epidermis was harvested and protein extracts run on Western blots and probed as above. For chronically-irradiated Hairless mice, 3–4 week old males were irradiated thrice weekly for a total weekly dose of 12.5 kJ/$m^2$ UVB (solar simulator, Oriel). At 72 days, PLX4720 treatment was started using drug-impregnated chow (Plexxikon) with vehicle chow in the control cohort.

## Soft agar assays

Following plating of bottom agar (0.6% Bacto Agar) with media and appropriate amounts of drug, 2500 to 10,000 cells per well (transformed WT, *Craf*−/− MEFs; transformed HaCaT SCR and TKD cells) were embedded in top agar (0.3%) and plated in 24-well plates. Control or drug-treated media was replaced every 48 hr for 4–6 weeks. The plates were stained with 1% crystal violet and colonies counted by bright-field microscopy.

## Statistical analysis

All data are represented as means ± SEM. All experiments were performed in triplicate at least. Student's t-test was used for comparison between two groups. $p \leq 0.05$ was considered significant.

# Acknowledgements

The authors acknowledge the assistance of Trellis Thompson in initial cell culture experiments, Sherie Mudd, Humaira Khan, Hafsa Ahmed, and Patricia Sheffield for histology, Nasser Kazimi and Omid

Tavana for assistance in UV radiation experiments, Haiching Ma, Sean Deacon, Gideon Bollag, Chao Zhang, Zhiqiang Wang, and R Eric Davis for experimental advice and reagents, and the South Campus Vivarium for mouse maintenance. KYT acknowledges Ronald P Rapini for departmental support as well as Tyler Jacks, Gordon B Mills, Patrick Hwu, and Jeffrey N Myers for critical discussions and mentorship. KYT acknowledges the funding support of DX Biosciences Cancer Research Fund, MD Anderson Cancer IRG Program, American Skin Association, Elsa U Pardee Foundation, institutional funds, and NCI CA16672 (FACS, Characterized Cell Line, and DNA Analysis Facility Cores). This paper is dedicated to the memories of Lee 'Sarge' Englet and Thomas C R Huang.

## Additional information

### Funding

| Funder | Grant reference number | Author |
|---|---|---|
| American Skin Association | | Kenneth Y Tsai |
| Elsa U Pardee Foundation | | Kenneth Y Tsai |
| University of Texas MD Anderson IRG Program | | Kenneth Y Tsai |
| DX Biosciences Cancer Research Fund | | Kenneth Y Tsai |
| National Cancer Institute | CA16672 | Kenneth Y Tsai |
| National Institutes of Health | GM059802, CA167505 | Kevin N Dalby |
| Welch Foundation | (F-1390) | Kevin N Dalby |

The funders had no role in study design, data collection and interpretation, or the decision to submit the work for publication.

### Author contributions

HV, SSO, KYT, Conception and design, Acquisition of data, Analysis and interpretation of data, Drafting or revising the article; GC, Acquisition of data, Analysis and interpretation of data, Drafting or revising the article; MLL, VC, DWD, CHA, MR, KNR, LRS, LD, Acquisition of data, Analysis and interpretation of data; SBF, Analysis and interpretation of data, Contributed unpublished essential data or reagents; DC, SEU, Conception and design, Analysis and interpretation of data; KE, MB, Provided critical advice and input on use of Craf−/− cells, Conception and design, Contributed unpublished essential data or reagents; RR, Provided critical advice and input on use of anti-ZAK antibody, Analysis and interpretation of data, Contributed unpublished essential data or reagents; JLC, Acquisition of data, Contributed unpublished essential data or reagents; KBK, AMC, MD, VGP, Identification of appropriate patient samples, Analysis and interpretation of data, Contributed unpublished essential data or reagents; KND, Conception and design, Analysis and interpretation of data, Contributed unpublished essential data or reagents; ERF, Conception and design, Analysis and interpretation of data, Drafting or revising the article

### Ethics

Human subjects: All human samples were archived tissue specimens made available for study through University of Texas MD Anderson Cancer Center IRB approved protocol LAB08-0750 (KYT). Informed consent specifically for these samples was not necessary due to the fact that they were de-identified, archived FFPE specimens.

Animal experimentation: This study was performed in strict accordance with the recommendations in the Guide for the Care and Use of Laboratory Animals of the National Institutes of Health. All of the animals were handled according to approved institutional animal care and use committee (IACUC) protocols (#06-09-06332) of the University of Texas MD Anderson Cancer Center. Every effort was made to minimize suffering.

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
