## [Decision Letter]

Thank you for sending your work entitled “BRAF inhibitors suppress apoptosis through off-target inhibition of JNK signaling” for consideration at *eLife*. Your article has been favorably evaluated by a Senior editor and 2 reviewers, one of whom is a member of our Board of Reviewing Editors.

The Reviewing editor and the other reviewer discussed their comments before we reached this decision, and the Reviewing editor has assembled the following comments to help you prepare a revised submission.

The authors investigate the contribution of non-ERK driven squamous cell cancers in melanoma patients treated with BRAF inhibitors. The clinical observation is that the incidence of SCC increases in patients treated with vemurafenib, which has been attributed to paradoxical increased RAF signaling due to the unrestrained activity of CRAF. Here they show that vemurafenib also inhibits a number of other serine threonine kinases including ZAK, which regulates JNK signaling. Expression of a ZAK mutant that cannot be inhibited by vemurafenib blocks this effect. The authors also demonstrate that dabrafenib has a different spectrum of kinase inhibition, suggesting that SCC may be less prevalent in patients treated with this inhibitor.

1) Overall the experiments in this manuscript are clearly presented and support the authors’ conclusions. They have used cell lines from human cancers and genetically engineered mice to great advantage. The use of many different cell lines in different parts of the manuscript is helpful but does lead to some question of whether all of the observations can be observed in at least 1 or 2 of the cell lines.

2) It has previously been reported (Davis et al. Nat. Biotech. (2011) 29, 1046) that ZAK, MKK4, BRK, SRMS etc bind to Raf inhibitors, including PLX-4720. The competitive binding assay presented for PLX-4720 appears to duplicate the previously published report. The authors should make it clear in the text what is new and what has been published previously.

3) Conclusions regarding the relative importance of ZAK, MKK4, and MAP4K5 are unclear. Moreover, the shRNA studies are poorly described. There are a number of questions concerning these studies:

a) The shRNA sequences are not described.

b) The ZAK and MAP4K5 antibody used for the shRNA analysis is not described.

c) The ZAK knockdown in Figure 3—figure supplement 1 appears clear, but the knockdown of MAP4K5 is unclear – this needs to be quantitated. Also, the extent of the MKK4 knockdown should be presented.

d) shRNA studies are described for ZAK and a triple KD of ZAK, MKK4, and MAP4K5. The effect of single knockdown of MKK4 or MAP4K5 using two different shRNA (each) should be shown. Without these data, conclusions concerning the relative roles of ZAK, MKK4, and MAP4K5 cannot be made.

4) ZAK is a poorly studied protein kinase. It would be helpful to the reader if information concerning ZAK tissue expression was presented.

5) SP600125 is an inhibitor of a large fraction of the kinome, including JNK. It is difficult to interpret any findings using this drug (see Phil Cohen’s publication in BJ).

---

## [Author Response]

*1) Overall the experiments in this manuscript are clearly presented and support the authors’ conclusions. They have used cell lines from human cancers and genetically engineered mice to great advantage. The use of many different cell lines in different parts of the manuscript is helpful but does lead to some question of whether all of the observations can be observed in at least 1 or 2 of the cell lines*.

We chose to replicate our results using ZAK knockdown in a different cell line (SRB1) by introducing two ZAK shRNA clones into these cells and demonstrating a similar diminution of JNK activity and apoptosis following UV exposure. These are now placed in Figure 3—figure supplement 3 and cited in the text in the Results section (“BRAFi suppress JNK activity through off-target inhibition of ZAK, MKK4, MAP4K5”).

*2) It has previously been reported (Davis et al. Nat. Biotech. (2011) 29, 1046) that ZAK, MKK4, BRK, SRMS etc bind to Raf inhibitors, including PLX-4720. The competitive binding assay presented for PLX-4720 appears to duplicate the previously published report. The authors should make it clear in the text what is new and what has been published previously*.

The binding assay used is the identical platform to that described in Davis et al. (2011), though their presented results were restricted to an estimate of *K*_*d*_ based upon one screening drug concentration of 10 micromolar. In our paper, we extended the panel of kinases (listed in Table 2), replicated some of the assays described in Davis et al., but included lower concentrations of drug to get a better estimated *K*_*d*_, and included vemurafenib, to verify that both drugs had similar effects on the kinases of interest. The other kinases we profiled were selected on the basis of literature supporting a role in signaling to JNK and ones that had been shown to bind or be inhibited by PLX4720/vemurafenib previously as controls. This is now more completely described in the main text in the Results section (“BRAFi suppress JNK activity through off-target inhibition of ZAK, MKK4, MAP4K5”).

*3) Conclusions regarding the relative importance of ZAK, MKK4, and MAP4K5 are unclear. Moreover, the shRNA studies are poorly described. There are a number of questions concerning these studies*:

*a) The shRNA sequences are not described*.

This is now clearly described in the Materials and methods section.

*b) The ZAK and MAP4K5 antibody used for the shRNA analysis is not described*.

This is now clearly described in the Materials and methods section.

*c) The ZAK knockdown in*
Figure 3—figure supplement 1
*appears clear, but the knockdown of MAP4K5 is unclear – this needs to be quantitated. Also, the extent of the MKK4 knockdown should be presented*.

This is now done using Image J densitometry controlled against the GAPDH loading control. The degree of knockdown is described in detail in Figure 3—figure supplement 1. In TKD cells, MKK4 is knocked down by 72.9% (lanes 1 vs. 3, Figure 3) and MAP4K5 by 54.6% (Figure 3—figure supplement 1).

*d) shRNA studies are described for ZAK and a triple KD of ZAK, MKK4, and MAP4K5. The effect of single knockdown of MKK4 or MAP4K5 using two different shRNA (each) should be shown. Without these data, conclusions concerning the relative roles of ZAK, MKK4, and MAP4K5 cannot be made*.

This is now shown in Figure 3—figure supplement 2. In short, single KD of either MAP4K5 or MKK4 only partially suppresses apoptosis and phospho-JNK induction. At least 71.9% knockdown was achieved for two MKK4 and MAP4K5 shRNA clones (Image J densitometry). The effect on apoptosis and phospho-JNK suppression is greater with MKK4 knockdown (up to 27.3%). This is however, completely expected, since MKK4 is important for full activation of JNK activity and ZAK has been shown to be upstream of MKK4 (Figures 2 and 3), thus implying that the effect of ZAK induction proceeds, at least in part, through MKK4. Consistent with this, we have also shown that ZAK knockdown and drug treatment suppresses phospho-MKK4 induction following irradiation (Figures 2 and 3). This is now discussed in the main text in the Discussion section.

*4) ZAK is a poorly studied protein kinase. It would be helpful to the reader if information concerning ZAK tissue expression was presented*.

This is now included in the main text in the Discussion section.

*5) SP600125 is an inhibitor of a large fraction of the kinome, including JNK. It is difficult to interpret any findings using this drug (see Phil Cohen’s publication in BJ)*.

We agree that SP600125 lacks the degree of specificity for JNK that would be desirable. We have therefore elected to remove these data from the paper given this problem. Parenthetically, we tried multiple JNK inhibitors in-vivo and were unable to consistently obtain pharmacodynamic evidence for specificity and in-vivo action.